# Comparative single-cell transcriptomic atlases of drosophilid brains suggest glial evolution during ecological adaptation

Daehan Lee [1,2*], Michael P. Shahandeh[1,3], Liliane Abuin[1], Richard Benton [1*]

1 Center for Integrative Genomics, Faculty of Biology and Medicine, University of Lausanne, Lausanne, Switzerland, 2 Department of Biological Sciences, College of Natural Sciences, Sungkyunkwan University, Suwon, Republic of Korea, 3 Department of Biology, Hofstra University, Hempstead, New York, United States of America

* Richard.Benton@unil.ch (RB); lee.daehan@skku.edu (DL)

## Abstract

To explore how brains change upon species evolution, we generated single-cell transcriptomic atlases of the central brains of three closely related but ecologically distinct drosophilids: the generalists *Drosophila melanogaster* and *Drosophila simulans*, and the noni fruit specialist *Drosophila sechellia*. The global cellular composition of these species' brains is well-conserved, but we predicted a few cell types with different frequencies, notably perineurial glia of the blood–brain barrier, which we validate *in vivo*. Gene expression analysis revealed that distinct cell types evolve at different rates and patterns, with glial populations exhibiting the greatest divergence between species. Compared to the *D. melanogaster* brain, cellular composition and gene expression patterns are more divergent in *D. sechellia* than in *D. simulans*—despite their similar phylogenetic distance from *D. melanogaster*—indicating that the specialization of *D. sechellia* is reflected in the structure and function of its brain. Expression changes in *D. sechellia* include several metabolic signaling genes, suggestive of adaptations to its novel source of nutrition. Additional single-cell transcriptomic analysis on *D. sechellia* revealed genes and cell types responsive to dietary supplement with noni, pointing to glia as sites for both physiological and genetic adaptation to this fruit. Our atlases represent the first comparative datasets for "whole" central brains and provide a comprehensive foundation for studying the evolvability of nervous systems in a well-defined phylogenetic and ecological framework.

## Introduction

Animal nervous systems contain hundreds to billions of cells. Encompassing neurons and glia, these cells can be categorized into a large number of different types—based upon developmental, anatomical, molecular, and functional properties [1,2]—with diverse roles. For example, neurons act in sensory detection, information processing,

**Data availability statement:** Raw sequencing data are archived in NCBI Gene Expression Omnibus (GEO) under accession code GSE247965. All bioinformatic and experiments datasets (including raw confocal images) for generating the figures and tables are available from Zenodo (https://zenodo.org/records/15016454). Processed single-cell transcriptomic atlases are available from SCope (https://scope.aertslab.org/#/DmelDsimDsec_brains).

**Funding:** DL is supported by the National Research Foundation of Korea (NRF) grants funded by the Ministry of Science and ICT under Project Numbers RS-2023-00211007 and RS-2023-00218602, and by the Ministry of Education under Project Number NRF-2019R1A6A1A10073079. Research in RB's laboratory is supported by the University of Lausanne, an ERC Advanced Grant (833548) and the Swiss National Science Foundation (310030B_185377). The funders had no role in study design, data collection and analysis, decision to publish, or preparation of the manuscript.

**Competing interests:** The authors have declared that no competing interests exist.

**Abbreviations:** ANOVA, analysis of variance; BBB, blood–brain barrier; DEG, differentially expressed gene; FDR, false discovery rate; GO, Gene Ontology; GPCR, G-protein-coupled receptor; Ms, Myosuppressin; OPN, olfactory projection neuron; PCA, principal component analysis; RPCA, reciprocal principal component analysis; snRNA-seq, single-nucleus RNA sequencing.

and locomotor control, while glia generally have support functions, including as structural and insulating scaffolds, in nutrient supply, and through removal of cellular debris and toxins [3,4]. The complement of cells in the nervous system of an extant species arises from ongoing evolutionary processes, where external selection pressures can lead to the emergence of new (or modified) cell types that fulfill functions conferring a fitness advantage (e.g., detecting a novel pertinent sensory stimulus, fine-tuning a motor action, or for modulating energy homeostasis). Conversely, cells whose function no longer contributes to organismal fitness might be lost (or repurposed). Understanding the genetic mechanisms and selective pressures underlying the gain, loss, and modification of cell types can reveal how and why nervous systems change over evolutionary timescales, as well as basic insights into the development and function of neural circuits [5,6].

Until recently, our understanding of nervous system evolution relied heavily on correlations of differences in the macroscopic (neuro)anatomy and behavior of different species [5–8], limiting our mechanistic (i.e., genetic) understanding of how evolutionary changes occur. Single-cell transcriptomic approaches have enormous potential to advance this knowledge, by enabling cataloging of neurons and glia and their molecular relationships in various species to suggest hypotheses for how and why divergence in cellular composition has occurred. For example, profiles of cerebellar output neurons in mice, chickens, and humans suggested that developmental duplications of subsets of these cells underlie the large expansion of the human cerebellum [9], while comparisons of cell type-specific transcriptomes in reptiles, amphibians, and other vertebrates have provided insights into both ancient cell types and mammalian-specific innovations of the cortex and other brain regions [10–12]. Such surveys can also relate gene and cell type evolution; for example, a comparison of hypothalamic neuronal populations in fish revealed that species-specific cell types are often characterized by the expression of species-specific gene paralogs [13].

A limitation of studying nervous system evolution in vertebrates is the very large number of cells in their brains: a sampling of >3 million cells from diverse sites in the human brain is a minuscule fraction of the estimated 100–200 billion cells of this organ [14], while a "whole"-brain scRNA-seq atlas of the mouse *Mus musculus* profiled ~7 million neurons [15], but this still represents only about 10% of the total [16]. Moreover, relating structural differences to ecologically relevant functional differences is often challenging. In this context, flies of the *Drosophila* genus define an excellent model clade for investigating nervous system evolution for several reasons. First, these species have relatively compact central brains, comprising ~43,000 cells (of which ~90% are neurons) in *Drosophila melanogaster* [17]. Second, *D. melanogaster* has been the focus of a wealth of molecular, anatomical, and functional studies relating brain structure to function over several decades [18–20], including the generation of single-cell transcriptomic atlases [21–23]. Third, different drosophilid species have adapted to diverse ecological niches—with different food sources, climate conditions, competitors, pathogens, and predators—and display numerous species-specific behaviors, including sensory responses to environmental stimuli and motor actions, such as courtship song production [24,25].

Amongst drosophilids, the trio of *D. melanogaster*, *Drosophila simulans*, and *Drosophila sechellia* present a particularly interesting set of species for comparative neurobiology. These species diverged from a common ancestor 3–5 million years ago, with *D. simulans* and *D. sechellia* diverging much more recently (100–250,000 years ago) [26,27] (Fig 1A). While *D. melanogaster* and *D. simulans* are cosmopolitan generalists, with overlapping geographic ranges and similar broad use of fermenting vegetal substrates for feeding and breeding, *D. sechellia* is endemic to the Seychelles archipelago and has evolved extreme ecological specialization, spending most or all of its life cycle exclusively on the "noni" fruit of the shrub *Morinda citrifolia* [28]. As noni fruit is toxic for other drosophilids, including *D. simulans* inhabiting the Seychelles [29,30], this niche specialization might alleviate interspecific competition, and possibly has essential nutritional benefits [31]. Commensurate with its unique ecology, *D. sechellia* displays many behaviors that are distinct from its generalist relatives, including olfactory and gustatory preferences, circadian plasticity, and certain reproductive behaviors [32–37]. Some of these behaviors have been linked to structural and/or functional changes in the peripheral nervous system. For example, several populations of olfactory sensory neurons display increased sensitivity to noni-derived odors in *D. sechellia* compared to *D. melanogaster* and *D. simulans*, due to coding mutations in specific olfactory receptors [32,33,38,39]. Furthermore, a number of olfactory sensory neuron populations are larger (or smaller) in *D. sechellia*, presumably due to changes in the developmental patterning of the olfactory organs [32,33,40–42], which can affect adaptation properties in the circuitry [42].

Despite these advances in defining causal, or at least correlative, relationships between peripheral sensory neuron changes and species-specific behaviors and ecologies of these flies, we know essentially nothing about if and how their central brains have evolved. Here, we performed comparative single-cell transcriptomic analyses of the central brains of this drosophilid trio to produce, to our knowledge, the first comparative whole central brain atlases in any species. Our results reveal conserved and divergent features of the cellular composition and gene expression patterns in these drosophilids' central brains as well as signals and patterns of brain evolution linked to niche specialization. This study provides a valuable dataset for future studies to investigate the genetic and cellular basis of brain evolution.

## Results

### Identification of diverse conserved cell types in drosophilid central brains

We generated central brain comparative single-cell transcriptomic atlases of *D. melanogaster*, *D. simulans*, and *D. sechellia* through single-nucleus RNA sequencing (snRNA-seq) (Fig 1A and 1B, Methods). In brief, we dissected the brains of 5-day-old, mated female adults cultured on standard food medium, and removed the optic lobes. All steps—tissue dissection, nuclear isolation, RNA extraction, library preparation, and sequencing—were performed in parallel for the three species for six biological replicates, each consisting of 20 brains per species. During the processing of snRNA-seq data with the Cell Ranger software (ver. 7.1.0) [43], sequence reads from *D. melanogaster*, *D. simulans*, and *D. sechellia* were mapped to their respective genomes. In total, the number of nuclei sequenced and analyzed per species (*D. melanogaster* = 49,830; *D. simulans* = 43,689; *D. sechellia* = 47,503) all exceed the estimated cell number in the central brain of *D. melanogaster* (~43,000 [17]). Our transcriptomic atlases therefore comprise ~1× cell coverage of their central brains. The median number of genes expressed per cell were 950, 923, and 861 for *D. melanogaster*, *D. simulans*, and *D. sechellia*, respectively, corresponding to 2001, 1841, and 1612 Unique Molecular Identifiers. Detailed metrics for the snRNA-seq results can be found in S1 Table.

To integrate and cluster the single-cell transcriptomic atlases of these three species, we identified 13,124 one-to-one-to-one gene orthologs among *D. melanogaster*, *D. simulans*, and *D. sechellia*. These genes were used to establish anchors for reciprocal principal component analysis (RPCA)-based integration across datasets from all three species (Fig 1C, Methods). Our dataset reproduced the previously described global structure of a brain single-cell transcriptome atlas in *D. melanogaster* [21], where cells are differentiated by their expression of developmental patterning genes (notably, *pros* and *Imp*) (S1A and S1B Fig). Glial cells (*repo*+) form several clusters, while neurotransmitter markers could

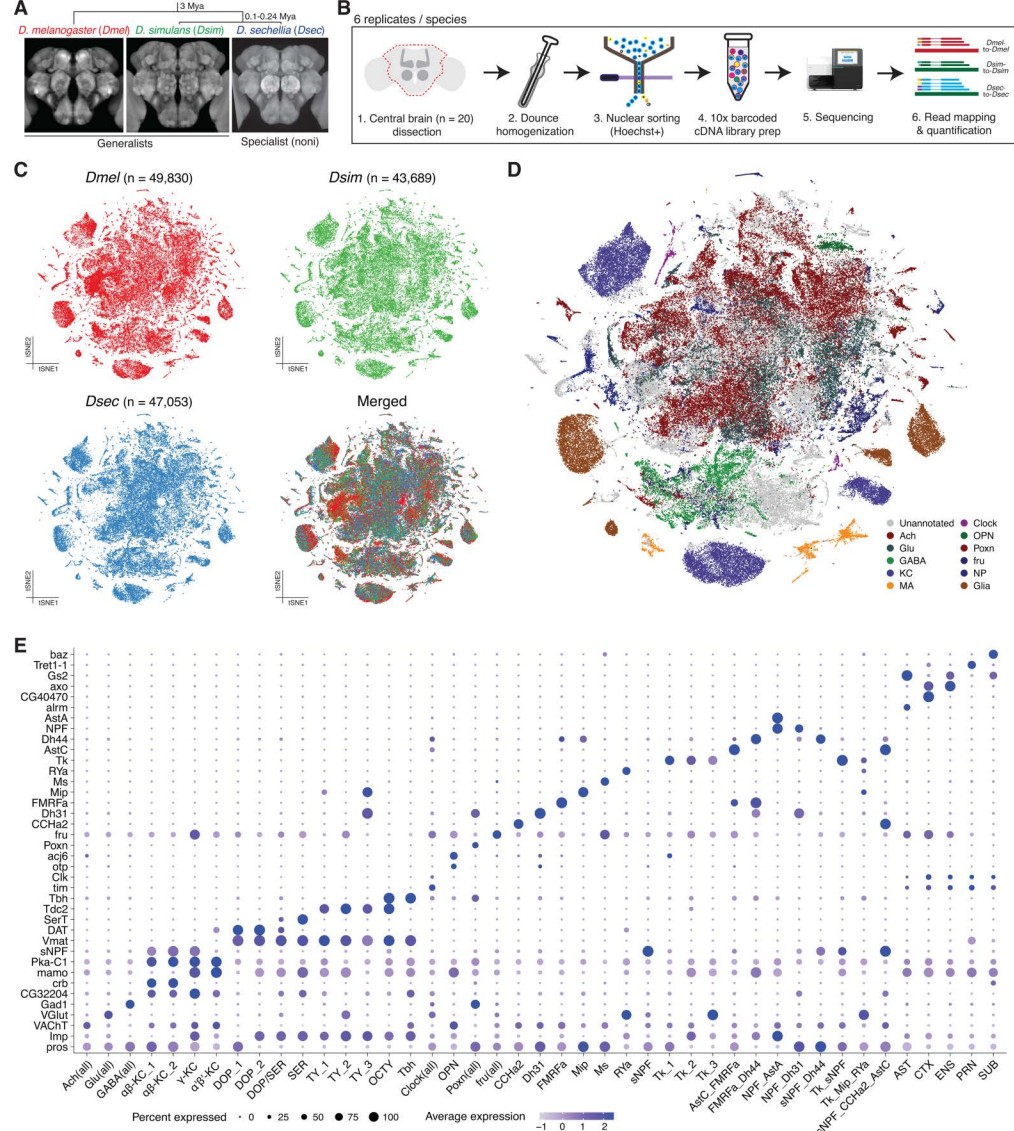

**Fig 1. Single-nucleus transcriptomic atlases of *D. melanogaster*, *D. simulans,* and *D. sechellia* central brains. (A)** Top: phylogeny of the drosophilid species studied in this work. Mya = Million years ago. Bottom: images of reference central brains for these species (all female; source: *D. melanogaster* [74]; *D. simulans* [75]; *D. sechellia* [76]). **(B)** Workflow of the single-nucleus RNA-sequencing of drosophilid central brains. **(C)** tSNE plots of *D. melanogaster* (red), *D. simulans* (green) and *D. sechellia* (blue) central brain cells from an integrated dataset after RPCA integration. In the bottom right plot, all cells from the three species are merged. **(D)** tSNE plot of the integrated and annotated datasets. Cells are colored by the 11 annotation groups. Unannotated cells are colored gray. **(E)** Dot plot summarizing the expression of genes used for the annotation of 107 cell types. Cholinergic, Glutamatergic, GABAergic, Clock, Poxn, and fru cell sub-types are merged into Ach (all), Glu (all), GABA (all), Clock (all), Poxn (all), and fru (all), respectively. Gene expression in this and other similar plots is shown as SCTransform-normalized Pearson residuals. The data underlying this figure can be found in https://zenodo.org/records/15016454.

distinguish GABAergic (*Gad1*+), monoaminergic (*Vmat*+), cholinergic (*VAChT*+), and glutamatergic (*VGlut*+) neurons (S1C and S1D Fig).

To characterize cell type diversity in the central brain, we performed iterative clustering and marker-based annotation (S2 and S3 Figs). After this iterative process, we classified clusters into eight groups (glia, Kenyon cells, monoaminergic neurons, Clock neurons, olfactory projection neurons [OPN], Poxn neurons, fru neurons, neuropeptidergic neurons);

the remaining clusters were grouped based on their neurotransmitter type (cholinergic, glutamatergic, GABAergic) (Fig 1D). For each group, we subclustered the cells and manually annotated subclusters based on marker gene expression, except the OPN that could not be further subclustered. In parallel, we disintegrated the dataset into three species-specific datasets to validate that clusters and their marker gene expression patterns are consistent across all datasets (Methods). Through this process, we identified 107 annotated cell types that are conserved across all three species (Fig 1E and S4 Fig). These cell types include five glial cell types, four Kenyon cell types, nine monoaminergic cell types, three Clock cell types, one OPN cell type, six Poxn cell types, 15 fru cell types, 18 neuropeptidergic cell types, 17 cholinergic cell types, 15 glutamatergic cell types, and 14 GABAergic cell types. In addition to previously characterized cell types [21,22], the larger size of our combined datasets provided power to identify novel, rare cell types. For example, we identified a cluster expressing the Myosuppressin (Ms) neuropeptide, which comprises less than 0.08% of central brain cells in all three species (S2 Table). These annotated cell types cover 63.8% of the total cells in the dataset, which were used for the downstream analysis. Our single-cell transcriptomic atlases show that the cellular composition of the central brain is globally conserved across *D. melanogaster*, *D. simulans*, and *D. sechellia*, consistent with their overall similarity in gross neuroanatomy (Fig 1A) [43]. Although we did not identify any examples of annotated cell types that are unique to any species, or absent in only one species, we cannot exclude that some examples might exist in the currently unannotated cells.

**Conserved gene expression patterns**

We first exploited our atlases to identify genes with conserved expression patterns—beyond the markers used in cell type annotation—which we reasoned would reveal the core molecules with essential roles in global brain organization and cellular functions. To identify such genes, we performed correlation analyses of cell type-specific gene expression levels across three pairwise species comparisons: *D. melanogaster-D. simulans* (*Dmel-Dsim*), *D. melanogaster-D. sechellia* (*Dmel-Dsec*), and *D. simulans-D. sechellia* (*Dsim-Dsec*). We focused on the 2,405 genes that are expressed in at least 5% of *D. melanogaster* central brain cells. The majority of these genes are broadly expressed in 5 to all 107 annotated cell types (on average, ~97 cell types)—where expression is counted if a gene is detected in at least 5% of the cells of a given type. For each of these genes, we computed the average expression level across 107 annotated cell types and compared these cell type-wide expression patterns across species, subsequently measuring the pairwise correlations (S5A Fig). This analysis identified 413 genes displaying strongly correlated expression patterns (Spearman's $p > 0.7$) across all three species pairings (S5B Fig and S3 Table). Gene Ontology (GO) term analysis revealed a notable enrichment in genes coding for membrane proteins, including adhesion molecules, ion channels, and G-protein-coupled receptors (GPCRs) (S5C Fig).

Next, we refined our analysis to genes that are more specifically expressed, positing that these are likely to govern cell type identity or underlie cell type-specific functions. Within each cell type, we identified genes that are consistently expressed in more than 30% of cells across all three species. Of these, we focused on the genes that are restricted to only 1–21 of the 107 annotated cell types. With this approach, we cataloged 925 genes exhibiting both conserved and specific expression patterns (S4 Table). This list of genes serves as a valuable resource for linking the functional roles of distinct brain cell types to their unique gene expression profiles. We note that although the chosen thresholds of gene expression specificity were arbitrary in this analysis, these datasets (S3 and S4 Tables) can be further mined with other thresholds.

The strong conservation of gene expression patterns across species implies the existence of shared gene regulatory mechanisms. While our profiling of mature adult brains is likely to limit our ability to identify important developmental genes, we reasoned that our datasets should still capture information on the expression patterns of terminal selector transcription factors, which establish and maintain the identity of post-mitotic neurons [44,45]. Of the 925 genes displaying specific and conserved expression patterns described above, 104 encode known or predicted transcription factors (S6 Fig and S4 Table). This group includes key developmental regulators, such as a POU domain transcription factor (*acj6*) and a paired-like homeobox transcription factor (*ey*), essential for the development of OPN and Kenyon cells, respectively)

[46–48]. These examples suggest the potential of other genes in this set to play critical roles in regulating cell type identity. For example, another paired-like homeobox transcription factor (*Ptx1*), which is expressed in a few cell types including Poxn_2 and several *fru+* cell types (S6 Fig), might be a terminal selector for these cell types, similar to its known role for enteroendocrine cell specification in the gut [49]. Furthermore, expression patterns of these transcription factors group cell types with shared functions and/or developmental origins (e.g., glia, Kenyon cells) (S6 Fig). Together, these analyses define a unique molecular fingerprint for each cell type's terminal identity and offer a resource for identifying previously unknown terminal selectors specific to each cell type (S4 Table).

## Divergence in the frequencies of homologous cell types

Having determined conserved types and molecular properties of likely homologous cell populations in the drosophilid trio, we next explored if and how these species' brains have diverged. Taking advantage of the statistical power afforded by having six biological replicates per species, we first analyzed interspecific variation in the representation of the 107 cell types. The majority of cell types displayed similar frequencies across the three species, including Kenyon cells, Clock neurons, and fru neurons (which play critical roles in learning and memory, circadian rhythms, and sexual behaviors, respectively) (Fig 2A and 2B and S7 Fig). From the comparison of *D. sechellia* with the other species, three cell types displayed significant variation (Fig 2A–C): there are significantly fewer perineurial glial cells (PRN)—which comprise the blood–brain barrier (BBB)—while one cholinergic cell type (Ach_1) and one Poxn-producing cell type (Poxn_1) are expanded. From the comparison of *D. simulans* with the other species, we found only one cell type, expressing the neuropeptide Tachykinin (Tk_1), that was significantly reduced (Fig 2A and 2B), while no cell types were reduced or expanded specifically in *D. melanogaster*.

To assess the observed differences in the cell composition of the *D. sechellia* brain atlases *in vivo*, we focused on PRNs, as the location of the Ach_1 and Poxn_1 subtypes of neurons amongst the much larger population of all Ach and Poxn neurons is unknown. We performed hybridization chain reaction fluorescence in situ hybridization (HCR FISH) experiments for *Tret1-1*, a specific marker for PRNs [50]. *Tret1-1* transcripts are broadly distributed across the PRN cell layer (likely because the transcripts are present throughout the cytoplasm of these large cells), which prevented direct quantification of PRN population size. However, as PRN *Tret1-1* RNA levels are not different between species (S8 Fig), we reasoned the total *Tret1-1* signal in the central brain should be a measure of PRN cell number. In *D. sechellia*, we found the global *Tret1-1* RNA signal is significantly lower, consistent with a reduction in PRN population size when compared to both *D. melanogaster* and *D. simulans* (Fig 2D and S9 Fig). To determine whether this reflects a reproducible interspecific difference, we performed *Tret1-1* RNA FISH in an independent strain of each of these species (not used in the snRNA-seq), observing the same lower signal in *D. sechellia* (Fig 2E). Finally, while PRNs were the only glial cell population with a significant decrease in representation in *D. sechellia*, we noted a similar trend for other types of glia (ensheathing [ENS] and cortex [CTX]) (S7 Fig), as well as all glia considered together (Fig 2B). We validated the latter prediction *in vivo* through immunofluorescence for the pan-glial nuclear marker, Repo, which revealed a lower number of glial cells in *D. sechellia* (Fig 2F and S9 Fig). These results suggest that our comparative single-cell atlases can provide reliable predictions for the interspecific variation in cellular composition of the central brains (as well as differences in cell population size within a species); in this context, we note that a snRNA-seq atlas of the *D. melanogaster* antenna that we generated in an independent study is remarkably representative of neuronal population size [51]. Our results suggest that the cellular composition of the central brain in *D. sechellia* is highly conserved with its generalist relatives, but a few cell types might have changed in number during ecological specialization.

## Transcriptomic divergence in homologous cell types

Given the global conservation of the types and frequencies of cells in these drosophilids' brains, we reasoned that phenotypic differences between these species might be reflected more prominently in the divergence of the transcriptomic profile of homologous cell types. We identified the 50 most abundantly expressed genes from each of the 107 cell types in

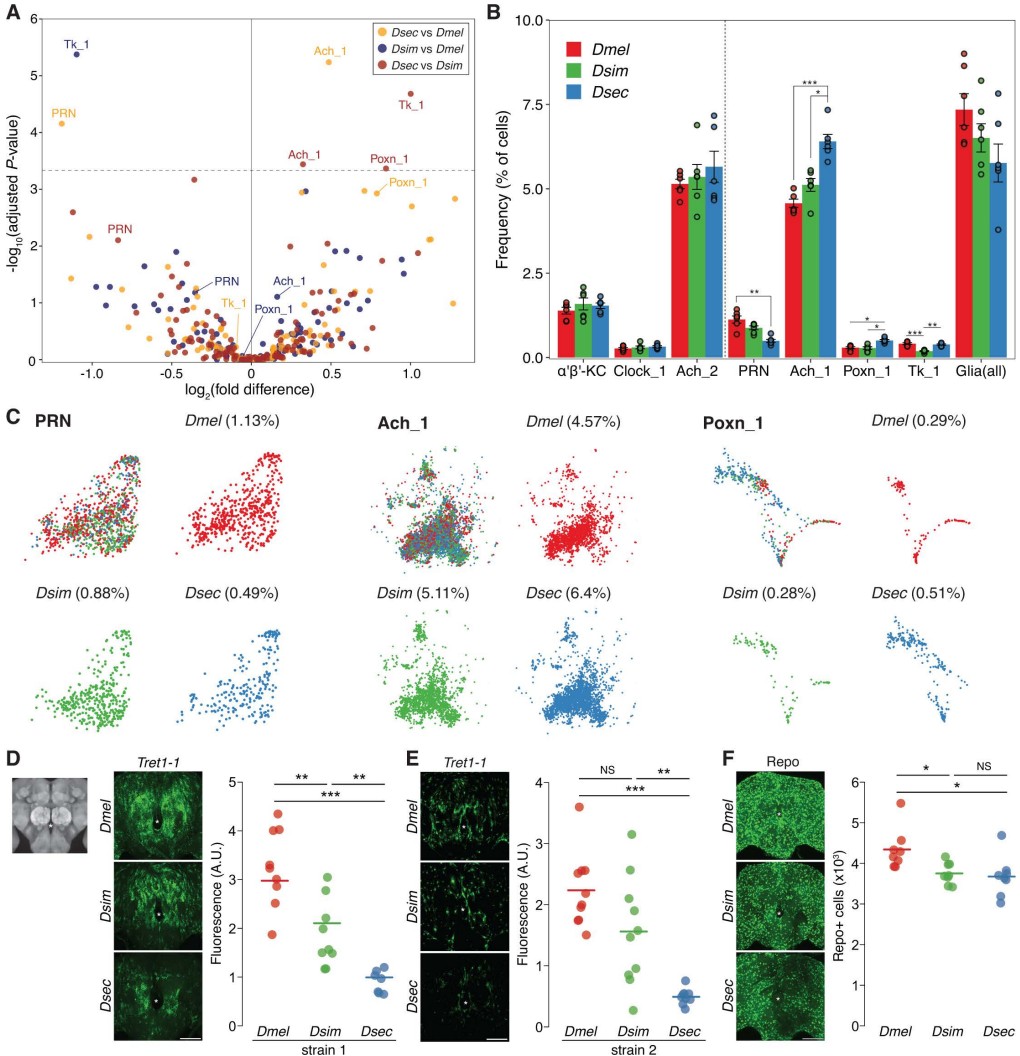

**Fig 2. Species divergence in the frequencies of central brain cell types. (A)** Volcano plot of the pairwise comparisons of cell type frequencies in the central brains of the three drosophilid species. The dashed horizontal line denotes the Bonferroni-corrected *P*-value threshold. **(B)** Bar plot illustrating selected cell type frequency comparisons. To the left of the dashed vertical line are three cell types with no significant differences across species. To the right, frequencies of four cell types exhibiting significant differences in at least one pairwise species comparison; far right, combined frequencies of all glial cell types. Each point corresponds to one of the six biological replicates. Statistical significance was calculated by one-way ANOVA with Tukey's HSD post-hoc test. *P*-values were adjusted using false discovery rate (FDR) for multiple comparisons. ***$P < 0.001$; **$P < 0.01$; *$P < 0.05$. **(C)** tSNE plots of PRN, Ach_1 and Poxn_1 populations. Cells are colored by their species of origin. The frequency of each cell type within its respective species is indicated in parentheses. **(D)** Left, representative images of *Tret1-1* HCR FISH on whole-mount brains of *Dmel*CS, *Dsim*04, and *Dsec*07 (see Methods for genotypes). The field of view of the brain in these and all other confocal images is shown on the far left (a cropped version of the *D. sechellia* reference brain from Fig 1A); the asterisk indicates where the esophagus would run, used as a central anatomical landmark. Right, quantifications of *Tret1-1* signals for each species. **(E)** Left, representative images of *Tret1-1* HCR FISH on *Dmel*OR, *Dsim*196, and *Dsec*28. Right, quantifications of *Tret1-1* signals for each species. **(F)** Left, representative images of Repo immunofluorescence on *Dmel*CS, *Dsim*04, and *Dsec*07. Right, quantifications of Repo+ cell counts for each species. The data underlying this figure can be found in https://zenodo.org/records/15016454.

*D. melanogaster* and examined the transcriptomic similarity between species for homologous cell types using correlation analysis (Fig 3A–3D). The similarity of gene expression profiles varies across cell types. For example, gene expression levels are highly similar in a Clock cell type (Clock_3) (*Dmel-Dsim*: $p = 0.670$, *Dmel-Dsec*: $p = 0.75$, *Dsim-Dsec*: $p = 0.86$) compared to OPN (*Dmel-Dsim*: $p = 0.43$, *Dmel-Dsec*: $p = 0.39$, *Dsim-Dsec*: $p = 0.61$). Notably, we found that many cell

types show reduced similarity in the *Dmel-Dsec* pair compared to the *Dmel-Dsim* pair (Fig 3B–3D), suggesting that gene expression profiles of these cell types have diverged more in the *D. sechellia* lineage than the *D. simulans* lineage.

To examine transcriptomic divergence at higher resolution, we characterized differentially expressed genes (DEGs, fold-change threshold >1.5, adjusted *P*-value < 0.05) within each cell type across the three species (*Dmel-Dsec*: n = 593, *Dmel-Dsim*: n = 487, *Dsim-Dsec*: n = 335) (S5 Table). Different cell types exhibited different DEGs, as illustrated for PRN and Poxn_1 cells (Fig 4A and 4B). Importantly, the vast majority of DEGs were identified as divergently expressed from only 1–3 cell types (Fig 4C). For example, among 593 DEGs from the *Dmel-Dsec* pair, 290 genes (48.9%) were identified from a single cell type, and 459 genes (77.4%) were identified from no more than three cell types, even though the majority of these genes are broadly expressed (on average, expressed in 79/107 cell types in *D. melanogaster*, minimum percent expression threshold: 5%). We next examined the cell type identity for each DEG (Fig 4D). Notably, in species comparisons with *D. sechellia* (*Dmel-Dsec* and *Dsim-Dsec*), four glial cell types (CTX, PRN, ENS, and astrocyte-like [AST]) consistently exhibited the highest number of DEGs. The higher DEG counts observed in glial cell types are not driven solely by their relatively large cell numbers (S10 Fig), suggesting that glial cell types display more divergent gene expression profiles than neurons in *D. sechellia*. Among the annotated neuronal cell types, a Poxn cell type (Poxn_3) expressing the neuropeptide Allostatin A (AstA) displayed the greatest number of DEGs in all three species comparisons. The observed cross-cell type and cross-species variation in transcriptomic divergence suggests that each cell type has undergone unique changes during the evolution of these species.

## Unique gene expression changes in *D. sechellia*

To investigate gene expression changes potentially related to the ecological specialization of *D. sechellia*, we analyzed the overlap of DEGs from different species comparisons. Sixty percent of DEGs were identified in more than one comparison (Fig 4E and 4F). For example, a set of 195 DEGs appeared in both *Dmel-Dsec* and *Dmel-Dsim*, but not in *Dsim-Dsec*; these shared DEGs might reflect expression changes originating either in the *D. melanogaster* lineage or in the common ancestor of *D. simulans* and *D. sechellia* (Fig 4G). Importantly, we observed 22% more DEGs from *Dmel-Dsec* (n = 593) than *Dmel-Dsim* (n = 487), indicating that the *D. sechellia* lineage has gained more gene expression changes (Fig 4E). When we inferred lineage-specific gene expression changes by investigating overlaps among DEGs (Methods), we found that the estimated number of *D. sechellia* DEGs (n = 96–258) far exceeded *D. simulans* DEGs (n = 48–152) (Fig 4G). Given that a similar number of DEGs would be expected if expression differences were solely due to divergence time ("neutral changes"), this observation suggests that the majority of DEGs in the *Dsim-Dsec* pair result from expression changes specific to the *D. sechellia* lineage.

We next focused on the 96 genes with expression changes specific to the *D. sechellia* lineage (S6 Table). Although we performed GO enrichment analysis on these 96 genes [52], the resulting terms were often supported by only one or two genes, limiting interpretability. Therefore, we relied primarily on manual curation to evaluate the putative or known functions of these genes. In neuronal cell types, we identified 54 *D. sechellia* DEGs including those encoding potassium channels (*SK*, *Slo2*), a vesicular glutamate transporter (*VGlut*), a calmodulin (*Cam*), an ecdysone-induced EF-hand (putative $Ca^{2+}$-binding) protein (*Eip63F-1*), an adenylate cyclase (*Ac3*) and at least four cell adhesion molecules for synapse assembly (*Cals*, *dpr1*, *dpr9*, and *side-IV*). One fru cell type (fru_5) and one Poxn cell type (Poxn_3) show the largest number of DEGs (n = 9), followed by an *NPF-/AstA*-positive cell type (NPF_AstA, n = 8) (Fig 5A).

Among glial cell types, we identified 42 *D. sechellia* DEGs in four glial cell types (CTX, AST, ENS, and PRN) (Fig 5A), consistent with the pronounced interspecific gene expression variation in glia (Fig 4D). These glial DEGs are enriched with various metabolic genes (Fig 5B and S11 Fig), including catalase (*Cat*), triglyceride lipase (*dob*), NAD(+) hydrolase (*sarm*), L-amino acid transmembrane transporter (CG4991), glycerophosphocholine phosphodiesterases (CG9394 and CG18135), and carbonic anhydrases (*CAH2* and *CAH3*). Intriguingly, for both of the latter two paralog pairs, we noted opposing expression changes in *D. sechellia*: CG9394 is downregulated but CG18135 is upregulated in AST (S11 Fig), and *CAH2* is upregulated and

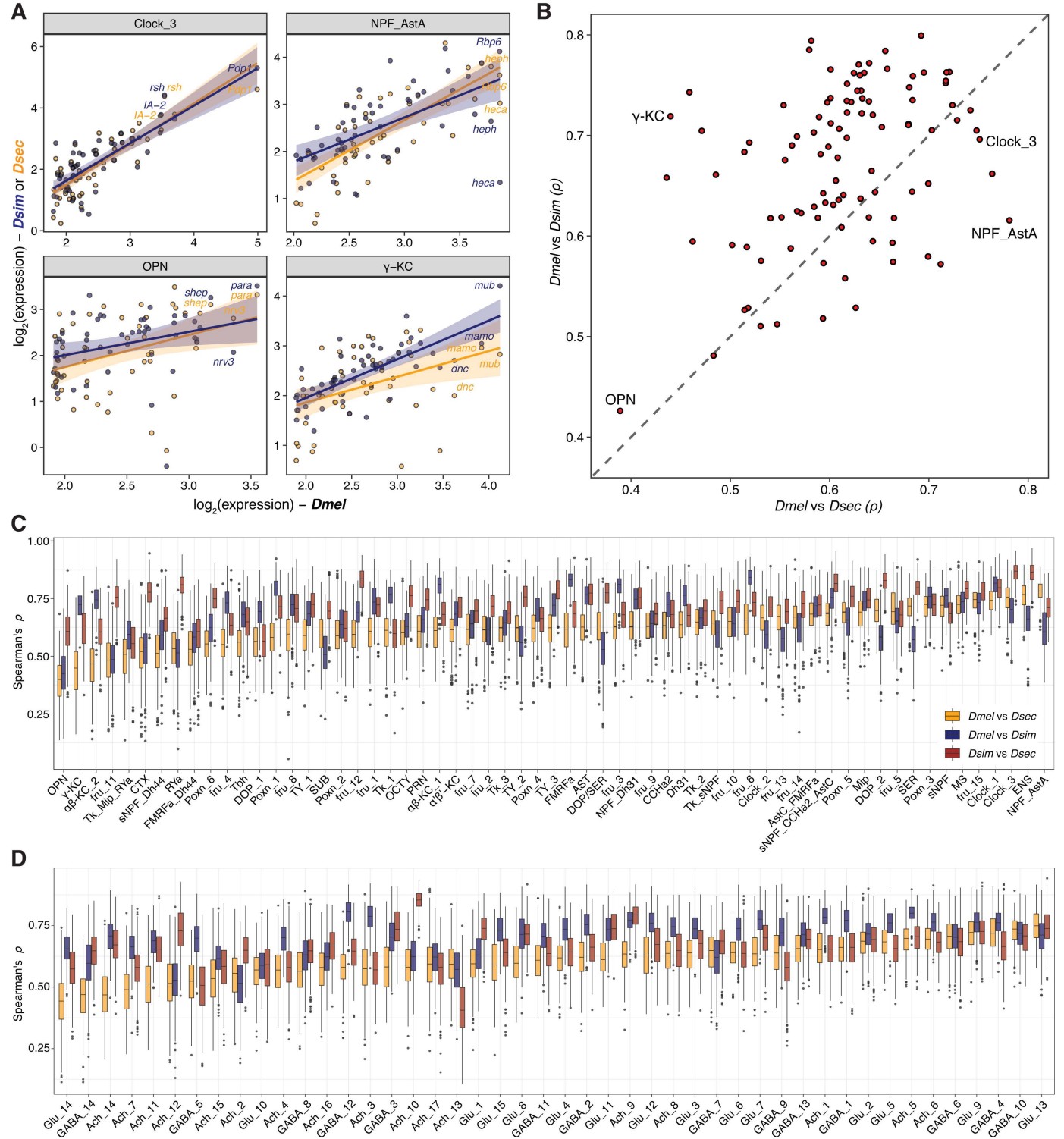

**Fig 3. Gene expression divergence of conserved central brain cell types across three drosophilid species. (A)** Scatter plots illustrating the interspecific gene expression variation across four distinct cell types: Clock_3, NPF_AstA (NPF-/AstA-positive cells), OPN (olfactory projection neurons), and γ-KC (Kenyon cells). The analysis features the 50 genes (each represented by a dot) with the highest expression levels in the pseudobulk transcriptome of *D. melanogaster* for each cell type. The plots display expression levels of these genes for three drosophilid species: *D. melanogaster* on the x-axis, and *D. simulans* (navy) and *D. sechellia* (orange) on the y-axis. Smoothed lines are linear model fits. **(B)** Scatter plot illustrating the transcriptomic

similarities among 107 annotated cell types between *D. melanogaster* and *D. sechellia* (x-axis), and *D. melanogaster* and *D. simulans* (y-axis). Each point represents an individual cell type. Spearman correlation coefficients (*p*) are derived from the expression level similarities of the top 50 highly expressed genes in the pseudobulk transcriptome of *D. melanogaster* for each respective cell type. **(C,D)** Tukey box plots presenting transcriptomic similarities among 107 annotated cell types (glia, Kenyon cells, monoaminergic neurons, Clock neurons, OPN, Poxn neurons, fru neurons and neuro-peptidergic neurons **(C)**; cholinergic, glutamatergic, and GABAergic neurons **(D)**) in pairwise comparisons of three drosophilid species. The Spearman correlation coefficients (*p*) are calculated based on the expression similarities of 30 randomly selected genes (from 400 permutations) among the top 50 highly expressed genes in *D. melanogaster*'s pseudobulk transcriptome for each cell type. The horizontal line in the middle of each box is the median; box edges denote the 25th and 75th quantiles of the data; and whiskers represent 1.5× the interquartile range. The data underlying this figure can be found in https://zenodo.org/records/15016454.

*CAH3* downregulated in PRN (Fig 5B). These observations imply potential gene expression dosage compensation between paralogs. From the glial DEGs, we also found the ecdysone-inducible gene E1 (*ImpE1*), which encodes a protein with low-density lipoprotein receptor type A repeats (Fig 5B). These results suggest that metabolic genes might have played a substantial role in the genetic adaptation to novel conditions in the *D. sechellia* lineage. To validate predicted gene expression differences *in vivo*, we performed HCR FISH for *CAH2* and *ImpE1*. For andc, we confirmed, in two independent strains, that gene expression levels are higher in *D. sechellia* compared to *D. melanogaster* or *D. simulans* (Fig 5C–5F and S12 Fig).

### Gene expression plasticity in the specialist brain

While these comparative atlases were intentionally generated from flies grown on the same food medium, we considered the possibility that the composition and transcriptome of the *D. sechellia* brain might be influenced by the presence of nutrients in noni. Supplementing food with noni juice paste greatly improved this species' fitness under laboratory conditions, as assessed by egg number and development to the pupal stage (Fig 6A). In parallel with the datasets described above, we also generated a central brain cell atlas from *D. sechellia* that had been reared on standard medium supplemented with noni paste. Comparisons of the snRNA-seq central brain transcriptomes of *D. sechellia* grown with or without noni supplement revealed essentially no effect on the cellular composition of the *D. sechellia* central brain (S13 Fig and S2 Table). Moreover, the examination of gene expression differences using the same threshold that yielded hundreds of interspecific DEGs, we identified only two genes: CG5151, which encodes a protein of unknown function, is more highly expressed in the brains of flies grown on noni; *Mkp3*, which encodes a mitogen-activated protein kinase phosphatase 3, is reduced upon noni supplement (Fig 6B and 6C). Interestingly, CG5151 is broadly expressed but differentially expressed specifically in PRN (S14 Fig), which is one of the cell types with the most significantly changed frequency and gene expression between species (Figs 2A–2E and 4D). To validate predicted gene expression plasticity upon noni supplement, we chose the most significant DEG, CG5151, for HCR FISH experiments. We confirmed that CG5151 is indeed upregulated, albeit at somewhat variable levels, in the central brains of two *D. sechellia* strains fed with noni paste (Fig 6D and S15 Fig).

When we lowered the fold-change threshold (to >1.2), we identified four more noni DEGs (Fig 6B). Three of these (*Cipc, SNF4Aγ*, and *Treh*) (S16 Fig) are differentially expressed in glial cell types, suggesting that glia are not only diverged in their gene expression between *D. sechellia* and two generalist *Drosophila* species but are also the most responsive to environmental conditions that drove the evolution of *D. sechellia* lineage. For example, expression of a trehalase (*Treh*) gene is higher in AST glia in *D. sechellia* than *D. melanogaster* or *D. simulans* when grown in the same standard medium; but *Treh* expression is decreased in *D. sechellia* grown on noni paste. The other noni DEG, *Vha100-1*, encoding a Vacuolar $H^+$ ATPase subunit, is differentially expressed in Poxn_3 neurons (S16 Fig), which is one of the neuronal types with the largest numbers of *D. sechellia* DEGs (Fig 5A).

### Discussion

Taking advantage of the small brains of closely related drosophilid species, our work provides an unprecedented, comprehensive view of the molecular and cellular conservation and divergence of animal brains in a well-defined phylogenetic

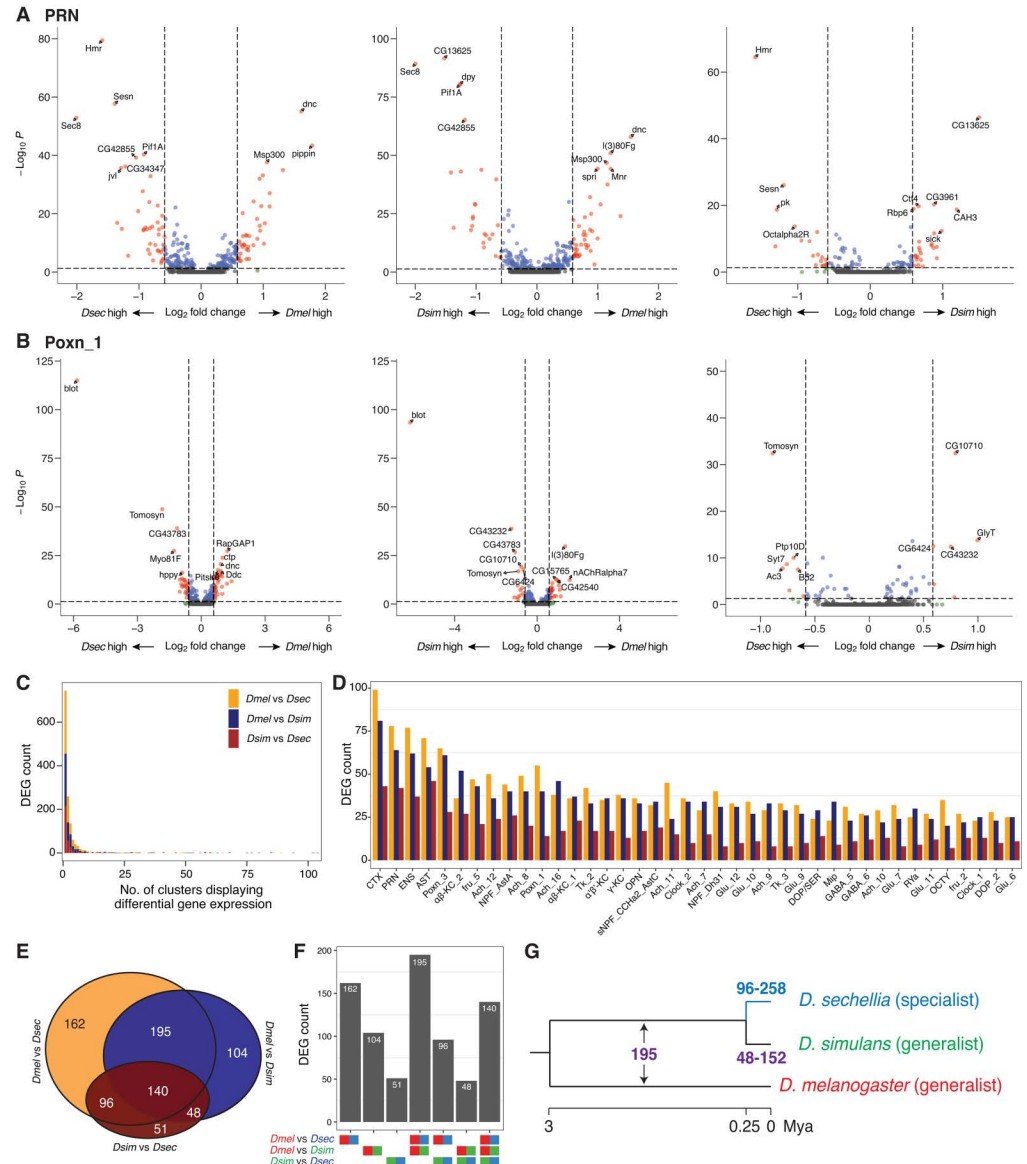

**Fig 4. Characterization of differentially expressed genes (DEGs) in conserved cell types across drosophilid central brains. (A,B)** Volcano plots showing DEGs (red) in two specific cell types, PRN **(A)** and Poxn_1 **(B)**, across three pairwise species comparisons. For comparisons with more than 10 DEGs, only the top 10 most significant genes are labeled. **(C)** Histogram displaying the frequency of DEGs sorted by the total number of cell types/clusters in which they are identified as DEGs. Stacked bars with distinctive colors represent DEGs identified in pairwise species comparisons. **(D)** A bar plot (color-coded as in **(C)**) showing the frequency of DEGs across the 40 cell types with the largest number of DEGs from all pairwise comparisons among the three species. Cell types are arranged in order of the total number of DEGs across three pairwise comparisons. **(E,F)** A Venn diagram **(E)** (color-coded as in **(C)**) and a bar plot **(F)** illustrating the intersections of DEGs identified in three pairwise species comparisons. **(G)** Hypothetical numbers of genes that underwent expression changes are shown across the lineages of *D. melanogaster*, *D. simulans*, and *D. sechellia*. Mya = Million years ago. The data underlying this figure can be found in https://zenodo.org/records/15016454.

and ecological framework. Previous studies on how the nervous system of *D. sechellia* differs from *D. melanogaster* and *D. simulans* have primarily focused on changes in peripheral chemosensory pathways [28,32–36,40,42], leaving knowledge of potential adaptations in the brain almost completely unexplored. Despite the very different ecology and behaviors of the equatorial island-endemic specialist *D. sechellia* from its cosmopolitan, generalist relatives, the overall

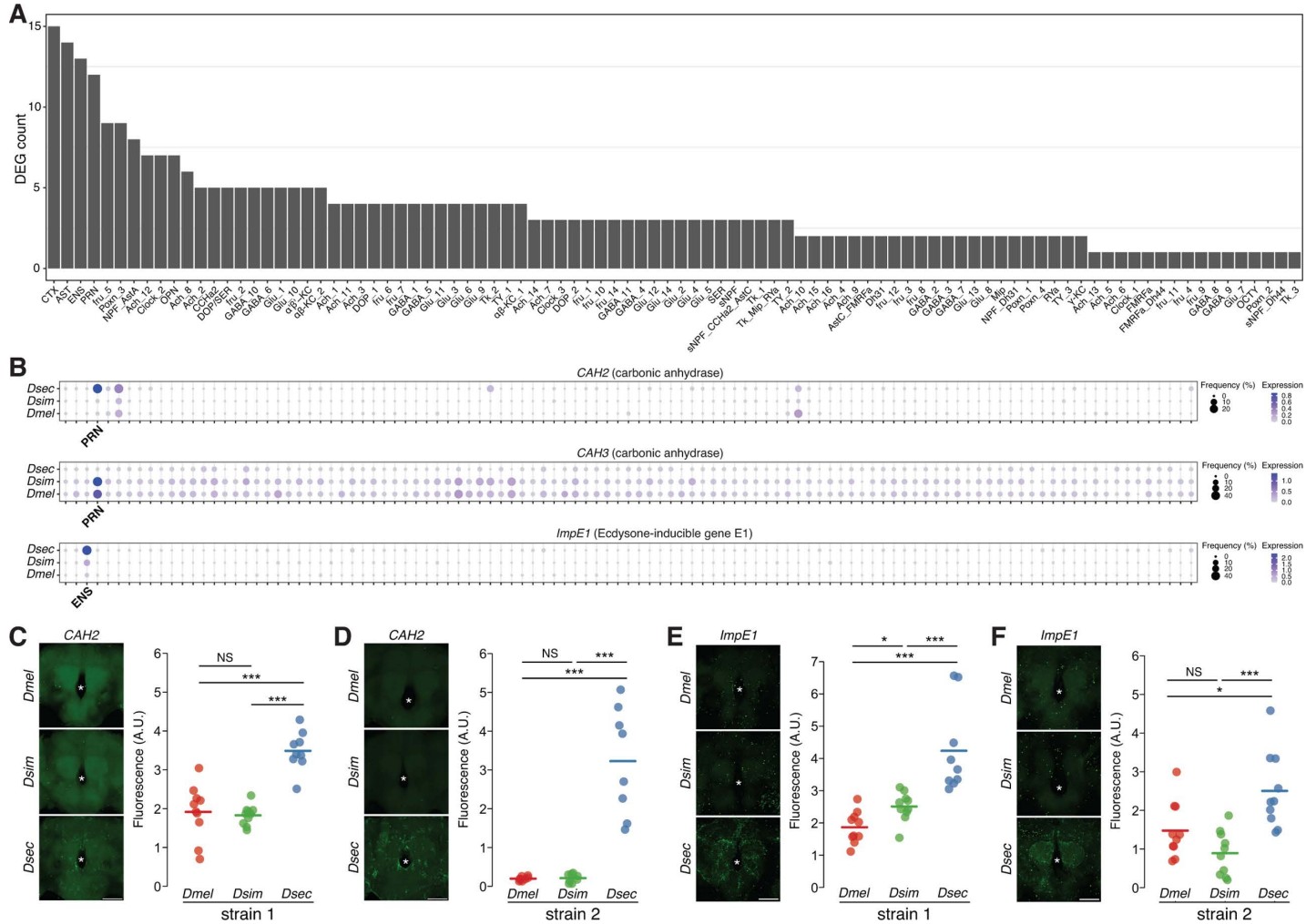

**Fig 5. Divergent gene expression in *D. sechellia*. (A)** Histogram of the frequency of *D. sechellia*-specific DEGs across 94 cell types, arranged by decreasing DEG number. The remaining 13 cell types with no DEGs are not shown. **(B)** Dot plots illustrating expression levels and frequencies of 3 *D. sechellia*-specific DEGs in *D. melanogaster*, *D. simulans*, and *D. sechellia* across 107 cell types. Cell types with significant differential expression are shown on the x-axis. **(C)** Left, representative images of *CAH2* HCR FISH on *Dmel*CS, *Dsim*04, and *Dsec*07. Right, quantifications of *CAH2* signals for each species. **(D)** Left, representative images of *CAH2* HCR FISH on *Dmel*OR, *Dsim*196, and *Dsec*28. Right, quantifications of *CAH2* signals for each species. **(E)** Left, representative images of *ImpE1* HCR FISH on *Dmel*CS, *Dsim*04, and *Dsec*07. Right, quantifications of *ImpE1* signals for each species. **(F)** Left, representative images of *ImpE1* HCR FISH on *Dmel*OR, *Dsim*196, and *Dsec*28. Right, quantifications of *ImpE1* signals for each species. The data underlying this figure can be found in https://zenodo.org/records/15016454.

brain architecture of these flies is highly conserved. We could classify 107 cell types in our atlases, though note that gene expression patterns alone are not necessarily sufficient to distinguish cell types within or between species [53], which will require developmental and anatomical analyses. Nevertheless, within the resolution of our clustering analyses, we did not detect any species-specific cell populations. Moreover, the frequencies of the vast majority of presumed homologous cell types are conserved. Our data provide empirical evidence to help begin answer the long-held question as to whether the sensory periphery is more evolvable than central brain regions [5]. The high conservation of central brain cell populations might reflect their more pleiotropic functions compared to individual sensory neuron populations, several of which have changed in size [32,33,40–42].

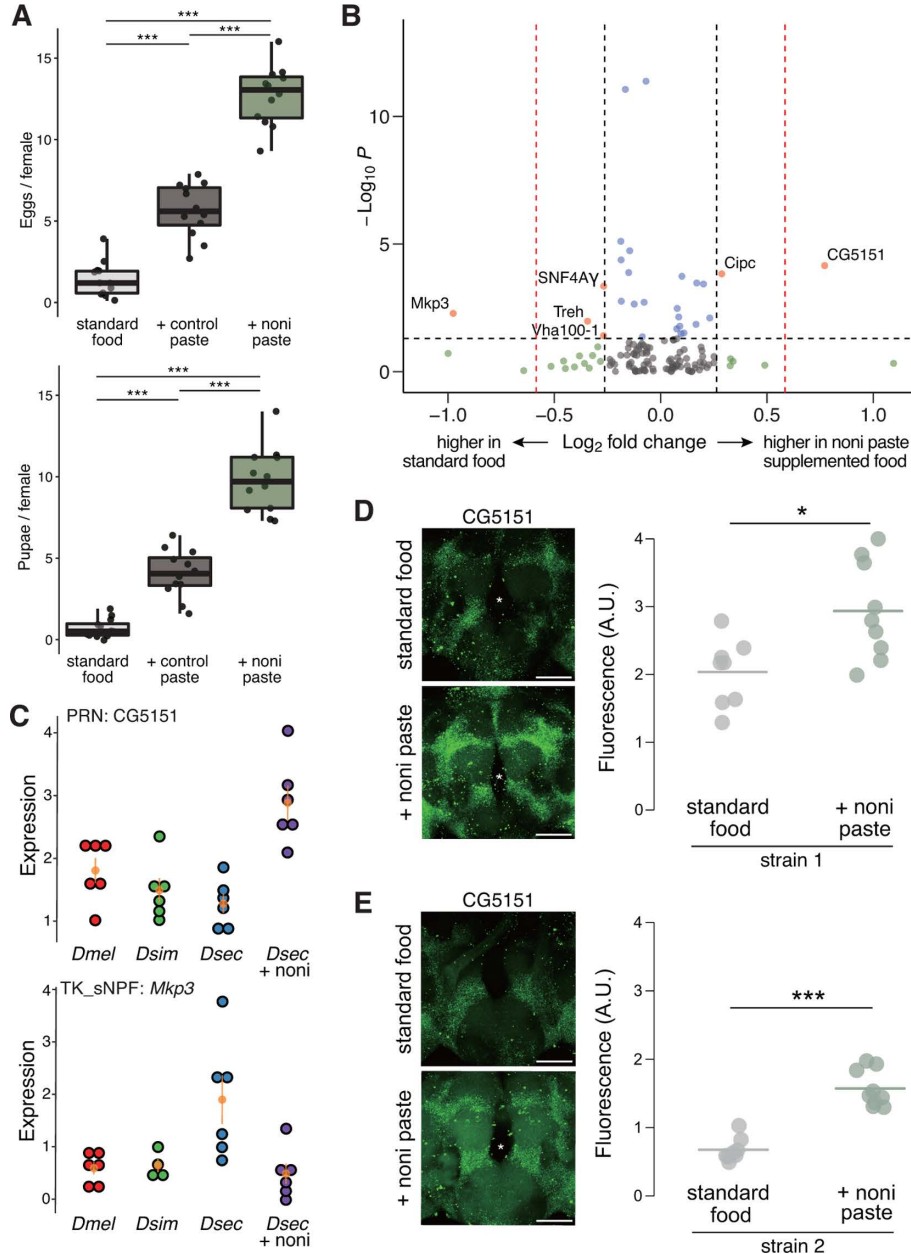

**Fig 6. Noni diet influences on *D. sechellia* brain gene expression. (A)** Tukey box plots for brood size (left) and pupal count (right) per mated *D. sechellia* female grown in three conditions. Each point corresponds to a biological replicate. The horizontal line in the middle of each box is the median; box edges denote the 25th and 75th quantiles of the data; and whiskers represent 1.5× the interquartile range. Statistical significance was calculated by one-way ANOVA with Tukey's HSD post-hoc test. ***$P < 0.001$. **(B)** Volcano plot highlighting differences in gene expression of *D. sechellia* central brains from animals grown without and with added noni paste. All cell type-specific DEG analyses are integrated into this single representation, and a reduced fold-change threshold (from 50% to 20%) is applied for the identification of DEGs. **(C)** Dot plots illustrating expression levels of CG5151 in PRN and *Mkp3* in TK_sNPF in the indicated species and growth conditions. **(D)** Left, representative images of CG5151 HCR FISH on *Dsec*07 grown under the indicated conditions. Right, quantifications of CG5151 signals. **(E)** Left, representative images of CG5151 HCR FISH on *Dsec*28 grown under the indicated conditions. Right, quantifications of CG5151 signals. The data underlying this figure can be found in https://zenodo.org/records/15016454.

In light of this conservation, consideration of the central brain cell populations that do differ between species is of interest. The most pronounced difference in *D. sechellia*'s central brain composition was found not in neurons but in a specific glial type, PRN, which forms a diffusion barrier around the nervous system as part of the BBB. In *D. melanogaster*, PRNs have an important role in sugar uptake into the brain [50]. Interestingly, noni fruit has a much lower carbohydrate (sugar):protein ratio than most other fruits, and *D. sechellia* cannot develop on carbohydrate-rich diets, apparently due to defects in carbohydrate-induced metabolic and gene expression changes [54]. We speculate that the reduction in PRNs in this species reflects a reduced requirement for sugar uptake by the brain. While it is unclear whether this loss is adaptive in *D. sechellia* (i.e., increasing fitness of this species in another, unknown, way), it might be a phenotype that constrains this species to this niche (i.e., representing an "evolutionary dead end" [55]). Only two neuron populations are expanded in *D. sechellia*—subtypes of cholinergic neurons and Poxn neurons—but as their roles even in *D. melanogaster* are unknown, it is difficult to interpret the significance of such expansions. Future exploration of these species differences *in vivo* might provide insights into as-yet poorly characterized central brain types in drosophilids.

Within homologous cell types, global patterns of gene expression—spanning those encoding known (or newly predicted) terminal transcription factors to signaling effectors, such as neurotransmitters/peptides—are, as expected, conserved. However, we observed many examples of species-specific divergence in gene expression, which was different in each cell type in terms of gene identity and magnitude of change. In particular, many broadly expressed genes differ in their expression between species only in a subset of specific cell types. Such properties reinforce the idea of cell types being independent evolutionary units [1]. Of interest is our observation that the majority of glial cell types display the greatest expression divergence. These properties might reflect lower selective pressures on glia to maintain precise structure and function compared to neurons that act within stringently defined circuitry. However, the enrichment of gene expression changes in glia might also reflect adaptive changes within these cell types, which have numerous known or presumed supportive roles in the brain, such as in homeostatic regulation and filtering of the external milieu for energy supply to neurons and waste removal [56].

It is important not to interpret every change in gene expression as indicative of adaptive evolution: after species divergence, neutral genetic polymorphisms accumulate, raising the possibility that gene expression also undergoes neutral evolution [57,58]. The drosophilid trio offers a useful model system to distinguish neutral and potentially non-neutral gene expression evolution. Under neutral conditions, the expectation is for *D. simulans* and *D. sechellia* to exhibit a similar degree of gene expression divergence when compared to *D. melanogaster*. However, our data revealed more pronounced shifts in brain gene expression in *D. sechellia*, suggesting that this lineage underwent non-neutral gene expression changes, possibly due to adaptation to its unique ecological niche. Glia stand out as the cell types displaying the most potential adaptive expression alterations, reinforcing the notion that they have an important contribution to brain evolution. Among the genes displaying altered expression patterns, those associated with metabolism are particularly enriched. The metabolic gene regulatory network might have undergone remodeling to adapt to the new nutritional conditions of the species' unique niche. An alternative hypothesis, however, is that *D. sechellia*'s smaller effective population size [59] reduces the efficacy of purifying selection, allowing neutral or slightly deleterious mutations to accumulate and greater expression divergence. Furthermore, the observed inverse expression changes in certain paralog pairs in *D. sechellia* (e.g., CG9394/CG18135 and *CAH2*/*CAH3*) suggest potential compensatory mechanisms that might further exacerbate divergence in gene expression profiles. Future analyses of signatures of selection on gene expression with broader phylogenetic sampling will be key to distinguishing adaptive and non-adaptive scenarios.

One challenge of comparing transcriptomes of ecologically diverse species is distinguishing between genetic and environmental effects on gene expression. To minimize environmental influences, our initial comparative atlases were of the drosophilid species grown on an identical food medium. However, as *D. sechellia* is thought to have metabolic defects [31,54], we also generated a brain atlas from flies grown with noni juice. Although this medium substantially increased *D. sechellia*'s survival and fecundity, there was remarkably little influence upon the cellular composition or gene expression

patterns in the brain of this species. These results support the idea that the interspecific differences observed are largely genetically determined. In contrast to our findings, a previous bulk RNA-seq analysis of whole flies identified several hundred DEGs upon noni exposure [60], suggesting that the brain might be "spared" [61] from nutrient-dependent gene expression changes compared to other tissues. Of the few observed environmental-dependent gene expression changes in the brain, the majority occur in various types of glia, emphasizing the plasticity of these cell types both over evolutionary timescales and in response to environmental changes. It is most noteworthy that the gene displaying the largest differential expression in *D. sechellia* cultured with and without noni (CG5151) marks the main cell type (PRN) displaying differential population size between species. These observations hint at the possibility that the cellular nodes in the brain that initially responded to the environmental presence of noni in the *D. sechellia* ancestor eventually have been reshaped by natural selection during species-specific niche adaptation [62].

Finally, we acknowledge that our atlases almost certainly do not capture the full extent of interspecific differences in gene expression. Although several lab-based studies of the peripheral olfactory system identified several olfactory sensory neuron populations that are increased or decreased in *D. sechellia* compared to the other species [32,33,40,42], it is possible that some interspecific variation in the brain remains cryptic under these controlled conditions. Additional experiments under more natural ecological contexts could reveal other brain-related adaptations. Furthermore, because our primary goal was to focus on species-specific differences and minimize variation from other biological factors, including sex, we analyzed only females in this study. The choice of sex was arbitrary, but in part guided by our focus on females in many of our prior investigations (e.g., noni attraction [33,42] and oviposition [34]). In the future, comparative studies on males will reveal sex-specific species differences (or those common to both sexes) in drosophilid central brains. In addition, as technical limitations of scRNA-seq currently constrain our ability to detect changes in rare cell types or genes with low expression, future deeper sequencing efforts and larger-scale single-cell atlases might reveal further differences. Beyond these possible extensions to the current work, broader sampling of diverse drosophilid species will be essential to situate changes in the *D. sechellia* lineage within a more expansive comparative framework. Such an integrated approach will advance our understanding of how neural circuits adapt to novel ecological niches and elucidate the fundamental mechanisms driving brain evolution.

## Methods

### *Drosophila* strains and culture

Flies used in this study were *D. melanogaster* (Canton-S (*Dmel*CS), Oregon-R (*Dmel*OR)), *D. simulans* (*Drosophila* Species Stock Center [DSSC] 14021-0251.004 (*Dsim*04), 14021-0251.196 (*Dsim*196)), and *D. sechellia* (DSSC 14021-0248.07 (*Dsec*07), 14021-0248.28 (*Dsec*28)), which were all cultured on standard wheat flour–yeast–fruit juice food. *Dmel*CS, *Dsim*04, and *Dsec*07 were used to generate the atlases. For the animals analyzed in Fig 6, *Dsec*07 was also grown on standard food supplemented on top with a paste of Formula 4–24 Instant *Drosophila* medium (blue, Carolina Biological Supply) mixed 1:5 weight:volume in noni juice (Raab Vitalfood).

### Single-nucleus RNA sequencing (snRNA-seq)

Thirty to fifty newly eclosed male and female flies were collected and placed together in food vials for mating, followed by sorting by sex on day 5. The central brains of 20 mated female flies were dissected and collected in 100 µl Schneider's medium, flash-frozen in liquid nitrogen, and stored at −80°C for nuclear extraction. Sample homogenization, single-nucleus suspension, and nuclear sorting were performed using the protocol described for the Fly Cell Atlas [23]. To obtain sequencing data from 10,000 nuclei, 15–20,000 nuclei were collected and loaded onto the Chromium Next GEM Chip (10x Genomics). Sequencing libraries were prepared with the Chromium Single Cell 3′ reagent kit v3.1 dual index, strictly following the manufacturer's recommendations. Libraries were quantified by a fluorimetric method, and their quality was assessed on a Fragment Analyzer (Agilent Technologies). Cluster generation was performed with 0.8–1.0 nM of

an equimolar pool from the resulting libraries using the Illumina HiSeq 3000/4000 PE Cluster Kit reagents. Sequencing was performed on the Illumina HiSeq 4000 using HiSeq 3000/4000 SBS Kit reagents according to 10× Genomics' recommendations (28 cycles read1, 8 cycles i7 index read, 8 cycles i5 index, and 91 cycles read2). Sequencing data were demultiplexed using the bcl2fastq2 Conversion Software (v2.20, Illumina). We performed six biological replicates for *D. melanogaster*, *D. simulans*, *D. sechellia* grown on standard food, and *D. sechellia* grown on noni-supplemented food (with each replicate for the four genotypes/conditions processed in parallel) resulting in snRNA-seq data from a total of 120 central brains per genotype/condition.

## Generation and integration of single-cell central brain transcriptomic atlases

Raw snRNA-seq data was first processed through Cell Ranger (v7.1.0, default parameters except --include-introns) [43]. The *D. melanogaster* reference genome and transcriptome from FlyBase (release 6.55) were used for all three species (*Dmel*-to-*Dmel*, *Dsim*-to-*Dmel*, *Dsec*-to-*Dmel*); sequence reads from *D. simulans* and *D. sechellia* were also processed with their own reference genomes (*D. simulans*: Prin_Dsim_3.1, *D. sechellia*: ASM438219v2) (*Dsim*-to-*Dsim*, *Dsec*-to-*Dsec*). Gene expression matrices from *Dsim*-to-*Dmel* and *Dsec*-to-*Dmel* processing were used for gene expression analysis (see below). All gene expression matrices were processed with SoupX (v1.6.2, default parameters) [63] to remove ambient RNA contamination. Decontaminated *Dmel*-to-*Dmel*, *Dsim*-to-*Dsim,* and *Dsec*-to-*Dsec* datasets were then normalized through SCT normalization [64], and putative doublets were filtered out using DoubletFinder (v2.0.4) [65]. Decontaminated and filtered datasets were subsetted for 13,124 one-to-one-to-one orthologs among reference genomes of *D. melanogaster*, *D. simulans*, and *D. sechellia*, which were inferred using Orthofinder [66] and manual curation of reciprocal best blastp hits; subsetted datasets were then integrated through the RPCA based integration method implemented in Seurat (v4.4.0, functions SelectIntegrationFeatures (nfeatures = 3000), PrepSCTIntegration, FindIntegrationAnchors (reference = *D. melanogaster*), and IntegrateData [67].

## Iterative subclustering

Principal component analysis (PCA) was performed on the integrated dataset; the first 50 PCs were used for clustering of cells in the integrated dataset into 36 clusters (functions FindNeighbors and FindClusters, default parameters except for resolution = 0.2). These clusters were further subjected to iterative subclustering. First, cells from each initial cluster were re-integrated at the cluster level. Subsequently, putative doublet clusters were identified and removed. Doublet clusters were defined as those exceeding 50% of the standard deviation of feature RNA counts, as these likely represent artifacts with an abnormally high transcript count. Following doublet filtering, the sizes of the resulting subclusters were evaluated. If a parent cluster did not segregate into at least two subclusters with frequencies >0.1% of the total cell population, iterative subclustering was terminated for that cluster. If there were at least two subclusters with frequencies >0.1%, the following steps were performed: (1) subclusters with frequencies <0.1% were reassigned to their parent cluster, (2) for subclusters with 0.1%–0.5% frequencies, the cells were assigned to the subcluster but not subjected to further subclustering, and (3) subclusters with >0.5% frequencies were subjected to an additional round of subclustering. This iterative subclustering process, summarized in S2 Fig, led to the definition of 240 clusters.

## Cell type annotation and quantification

Marker genes for 240 clusters obtained through iterative subclustering were identified using the FindMarkers function in Seurat (only.pos = TRUE, min.pct = 0.15, logfc.threshold = 0.25, test.use = "MAST"). These marker genes were used to classify the clusters. Cluster identities were primarily determined by comparing these marker genes with those identified in previous studies for *D. melanogaster* [21,22]. Any clusters and subclusters that remained unannotated following this step were then identified based on marker genes involved in neurotransmission and neuromodulation [68]. The annotation

process involved sequentially assessing the expression of marker genes associated with specific cell types (S3 Fig), including glial cells, Kenyon cells, monoaminergic neurons, Clock neurons, OPN, Poxn neurons, and fru neurons. For the remaining unclassified clusters, neuropeptide gene expression was examined. Fourteen neuropeptide genes expressed in fewer than 5% of clusters were identified, and clusters expressing these genes were classified as neuropeptidergic neurons. The remaining clusters were further categorized into three groups based on their neurotransmitter production: cholinergic, glutamatergic, or GABAergic. Following this classification, same class clusters were grouped, and cells within each group were re-integrated and subsequently segregated into individual species datasets. PCA and clustering were re-performed for these datasets. Gene expression profiles were then analyzed to identify marker genes specific to the resulting clusters. Only clusters with clear and distinct marker gene signatures were retained and annotated as specific cell types. With this workflow (S2 Fig), we obtained 107 annotated cell types. 63.8% cells belong to these annotated cell types; we only included these cells and annotated cell clusters for the downstream analysis. The frequency of each cell type was determined by quantifying the proportion of cells assigned to a specific cell type relative to the total number of cells in each scRNA-seq experiment, with six replicates per species/condition. The significance of interspecific variation in cell type frequency was calculated by a one-way analysis of variance (ANOVA) test, followed by Tukey's post hoc test for multiple comparisons.

### Conserved gene expression analysis

To mitigate artifacts stemming from the use of different reference genomes, we selected genes that exhibited similar expression levels when aligned to multiple genomes. Specifically, reads from *D. simulans* and *D. sechellia* were aligned to both the *D. melanogaster* genome and their own genomes. Subsequently, the percent expression for every gene was calculated across all cells and ranks between the two alignments (one to the *D. melanogaster* genome and the other to their own genomes) were compared. Of 7,926 genes that exhibited less than a 5% rank difference for both *D. simulans* and *D. sechellia* samples, we retained 2,405 genes that were expressed in at least 5% of *D. melanogaster* central brain cells for the analysis. The AverageExpression function in Seurat was employed to calculate the average gene expression of these selected genes for each cell type. Expression levels of these genes across 107 cell types were then compared across species, with the similarity between species being assessed through correlation analysis using Spearman's *p*. FlyEnrichr ([https://maayanlab.cloud/FlyEnrichr/](https://maayanlab.cloud/FlyEnrichr/)) was used for the GO analysis on 413 genes with conserved expression patterns [52,69].

### Transcriptomic comparisons in homologous cell types

Before comparing gene expression between homologous cell types, we selected genes with reliable expression level information for every cell type by excluding genes whose expression levels in *D. simulans* or *D. sechellia* showed discrepancies (5% rank difference) when their transcripts were aligned to either *D. melanogaster* or their own reference genomes. With these criteria, we omitted 1146 genes (13.8% of analyzed genes). Subsequently, we identified the top 50 most abundantly expressed genes within each of the 107 annotated cell types in *D. melanogaster*. Transcriptomic similarities between species for homologous cell types were assessed through correlation analysis.

### Differential gene expression analysis

For cell type-specific DEG analysis, cells were subsetted based on their cell type membership and grouped by species and growth conditions (four groups: *Dmel*, *Dsim*, *Dsec,* and *DsecNoni*). DEGs were identified using the FindMarkers function in Seurat (test.use = "MAST", min.pct = 0.05). To reduce false discoveries of DEGs that might arise from discrepancies or errors in the genome annotation of different species, sequence reads from *D. simulans* and *D. sechellia* were mapped to their own reference genome and to that of *D. melanogaster*. A gene was only designated as differentially

expressed if the mapping of the gene to both genomes consistently indicated differential expression (adjusted $P$-value < 0.05, fold-difference >50%) between the species or condition. For gene expression plasticity analysis, a lower fold-difference threshold (>20%) was also examined. The gene *Hr38* was excluded from the plasticity analysis because differential expression of this gene was driven by a single replicate, leading to unreliable results.

### Lineage-specific gene expression changes

To infer gene expression changes that occurred after the divergence of *D. simulans* and *D. sechellia* lineages, two methodologies were implemented. The initial approach comprised of identifying non-overlapping DEGs between *Dmel-Dsim* DEGs and *Dmel-Dsec* DEGs, under the assumption that these DEGs do not encompass gene expression differences between the *Dmel* lineage and *Dsim*/*Dsec* lineage before their divergence. This method established the highest counts for *D. simulans* and *D. sechellia* DEGs. The second approach identified the intersection between DEGs of *Dsim-Dsec* and *Dmel-Dsim* or *Dmel-Dsec*. This approach assumed that any *Dsim* or *Dsec* lineage-specific changes would be captured by overlap between *Dsim-Dsec* and *Dmel-Dsim* (*D. simulans* DEGs) or *Dsim-Dsec* and *Dmel-Dsec* (*D. sechellia* DEGs), respectively. This approach established minimum counts for *D. simulans* and *D. sechellia* DEGs.

### HCR FISH

Probe libraries for HCR FISH were synthesized by Molecular Instruments, based on the *D. melanogaster* gene sequences (sequence identity was >96% for all genes across all three species). 3–5-day-old females were fixed in 4% paraformaldehyde for 2 h prior to brain dissection and HCR FISH was performed essentially as described [70]. Brains were imaged using an inverted confocal microscope (Zeiss LSM 880) equipped with a 40× objective, using fixed settings to maximize the comparability of images. The 40× objective permitted a region of interest containing the entire midbrain but excluding the optic lobes to facilitate direct comparison to the snRNA-seq data (which did not include the optic lobes). Images were captured using Microsoft ZEN 2.3 SP1 software. We quantified total fluorescence in the midbrain as a proxy for RNA abundance; images were processed in Fiji by first creating a maximum-intensity projection z-stack of the midbrain. Relative fluorescence was calculated as the integrated density of pixel intensity. We compared total fluorescence between strains using a Wilcoxon rank-sum test followed by *post hoc* correction for multiple tests. To ensure repeatability of the expression differences we observed, we replicated each experiment twice in our focal strains (*Dmel*CS, *Dsim*04, and *Dsec*07—"strain 1"), as well as another set of wild-type strains of the same species (*Dmel*OR, *Dsim*196, and *Dsec*28—"strain 2"). We did not compare between, or pool, replicates because these flies were dissected, labeled, and imaged separately. Beyond the examples illustrated in this work, transcripts for several other genes were not reliably detected in any species, likely because they are expressed below the detection threshold of this methodology.

### Immunofluorescence

To compare total glial cell numbers between species, we performed immunofluorescence for the glial cell marker Repo [71]. 3−5-day-old brains of female flies were fixed, dissected, and stained essentially as described [72], using mouse-anti-Repo (1:20) (8D12) (Developmental Studies Hybridoma Bank AB_528448) and Cy5 goat anti-mouse secondary antibody (1:500) (Molecular Probes, Jackson Immunoresearch 115-175-166). Images of brains were obtained using an inverted confocal microscope (Zeiss LSM 880) equipped with a 40× objective, using fixed settings. Here, to best approximate the snRNA sequencing protocol, the optic lobes were masked before quantification. Images were processed in Fiji and Repo-positive cells in the midbrain were counted using the Fiji macro, RS-FISH [73]. No signal was detected outside the brain. The Repo-positive cell counts were compared between strains using a Wilcoxon rank-sum test followed by post hoc correction for multiple tests. We repeated this experiment twice to ensure replicability of the overall pattern. We did not pool, or compare between, replicates because these flies were dissected, stained, and imaged separately.

## Fitness assay

The fitness of *D. sechellia* was measured under three conditions: standard food, standard food supplemented with Formula 4–24 Instant *Drosophila* Medium blue (Carolina Biological Supply) that was hydrated with distilled water at a 1:5 weight-to-volume ratio ("control paste" in Fig 6A), or with noni paste (described above; "noni paste" in Fig 6A). For each condition, 10 mated, 5-day-old females were introduced into a culture vial. Brood sizes were measured after 24 h, followed by counting the number of pupae on day 8–9.

## Supporting information

**S1 Table. Summary of the metrics from the snRNA-seq experiments.**
(XLSX)

**S2 Table. Frequencies of 107 cell types in *D. melanogaster*, *D. simulans*, and *D. sechellia.***
(XLSX)

**S3 Table. 413 broadly expressed genes (>5% of central brain cells in *D. melanogaster*) with conserved expression patterns (Spearman's *p* > 0.7).**
(XLSX)

**S4 Table. 925 genes with specific expression (<22/107 cell types) and conserved expression patterns (expressed in >30% of cells across all three species).** The subset encoding predicted/known transcription factors (TFs) (GO:0000981) are shown separately on the second worksheet.
(XLSX)

**S5 Table. Differentially expressed genes identified from pairwise comparisons of three drosophilid species in a cell type-specific manner.**
(XLSX)

**S6 Table. 96 genes with putative expression changes specific to the *D. sechellia* lineage.**
(XLSX)

**S1 Fig. Integrated single-cell transcriptomic atlases of *D. melanogaster*, *D. simulans*, and *D. sechellia*. (A,B)** tSNE **(A)** and UMAP **(B)** plots of the integrated dataset after RPCA integration of single-cell transcriptomes of *D. melanogaster*, *D. simulans*, and *D. sechellia* central brain cells colored by the expression of *pros* or *Imp*. **(C,D)** tSNE **(C)** and UMAP **(D)** plots with cells colored by their predicted molecular identities. Neurons are identified through markers for neurotransmitter production, color-coded according to GABAergic (*Gad1*), monoaminergic (*Vmat*), glutamatergic (*Vglut*), and cholinergic (*VAChT*) designations. Glia, labeled by *repo* expression, are colored pink. Cells failing to meet expression thresholds for these markers are colored gray. The data underlying this figure can be found in https://zenodo.org/records/15016454.
(EPS)

**S2 Fig. Workflow for snRNA-seq data processing. (A)** Workflow of the generation of comparative single-cell transcriptomic atlases (see Methods). **(B)** Workflow of the iterative subclustering (ISub) (see Methods).
(EPS)

**S3 Fig. Marker gene-based cluster classification.** Workflow for classifying post-ISub clusters into 11 groups (glia, Kenyon cells, monoaminergic neurons, Clock neurons, olfactory projection neurons (OPN), Poxn neurons, fru neurons, neuropeptidergic neurons, cholinergic neurons, glutamatergic neurons, and GABAergic neurons).
(EPS)

**S4 Fig. Cell type annotations in central brains of *D. melanogaster*, *D. simulans*, and *D. sechellia*. (A)** Glial cell types. Left, UMAP plots of the glial cell type annotation for integrated and disintegrated single-cell transcriptomic atlases of *D. melanogaster*, *D. simulans*, and *D. sechellia*. Right, expression levels of marker genes distinguishing the glial cell types. **(B)** Kenyon cell types. Left, UMAP plots of the Kenyon cell type annotation for integrated and disintegrated single-cell transcriptomic atlases of *D. melanogaster*, *D. simulans*, and *D. sechellia*. Right, expression levels of marker genes distinguishing the Kenyon cell types. **(C)** Monoaminergic cell types. Left, UMAP plots of the monoaminergic cell type annotation for integrated and disintegrated single-cell transcriptomic atlases of *D. melanogaster*, *D. simulans*, and *D. sechellia*. Right, expression levels of marker genes distinguishing the monoaminergic cell types. **(D)** Clock cell types. Left, UMAP plots of the Clock cell type annotation for integrated and disintegrated single-cell transcriptomic atlases of *D. melanogaster*, *D. simulans*, and *D. sechellia*. Right, expression levels of marker genes distinguishing the Clock cell types. **(E)** Poxn cell types. Left, UMAP plots of the Poxn cell type annotation for integrated and disintegrated single-cell transcriptomic atlases of *D. melanogaster*, *D. simulans*, and *D. sechellia*. Right, expression levels of marker genes distinguishing the Poxn cell types. **(F)** fru cell types. Left, UMAP plots of the fru cell type annotation for integrated and disintegrated single-cell transcriptomic atlases of *D. melanogaster*, *D. simulans*, and *D. sechellia*. Right, expression levels of marker genes distinguishing the fru cell types. **(G)** Neuropeptidergic cell types. Left, UMAP plots of the neuropeptidergic cell type annotation for integrated and disintegrated single-cell transcriptomic atlases of *D. melanogaster*, *D. simulans*, and *D. sechellia*. Right, expression levels of marker genes distinguishing the neuropeptidergic cell types. **(H)** Cholinergic cell types. Left, UMAP plots of the cholinergic cell type annotation for integrated and disintegrated single-cell transcriptomic atlases of *D. melanogaster*, *D. simulans*, and *D. sechellia*. Right, expression levels of marker genes distinguishing the cholinergic cell types. **(I)** Glutamatergic cell types. Left, UMAP plots of the glutamatergic cell type annotation for integrated and disintegrated single-cell transcriptomic atlases of *D. melanogaster*, *D. simulans*, and *D. sechellia*. Right, expression levels of marker genes distinguishing the glutamatergic cell types. **(J)** GABAergic cell types. Left, UMAP plots of the GABAergic cell type annotation for integrated and disintegrated single-cell transcriptomic atlases of *D. melanogaster*, *D. simulans*, and *D. sechellia*. Right, expression levels of marker genes distinguishing the GABAergic cell types. The data underlying this figure can be found in https://zenodo.org/records/15016454.
(PDF)

**S5 Fig. Genes with conserved expression patterns across the central brains of *D. melanogaster*, *D. simulans*, and *D. sechellia*. (A)** Scatter plots illustrating the expression pattern comparisons of two genes, *bru3* (top) and *gw* (bottom), between *D. melanogaster* and *D. sechellia*. Each point corresponds to one of the 107 cell types. Average expression levels of *D. melanogaster* are shown on the x-axis, and *D. sechellia* expression levels are shown on the y-axis. Smoothed lines depict generalized linear model fits. **(B)** Top: two-dimensional density plots of gene expression pattern similarity for three pairwise comparisons. Each point in the plot corresponds to one of 2,405 analyzed genes. The y-axis shows the expression pattern similarity, which is estimated by the correlation value (Spearman's $p$). Bottom: histograms of gene count with highly conserved expression patterns ($p > 0.7$) are shown. The x-axis shows the percentage of central brain cells expressing each gene. **(C)** GO enrichment analysis for 413 genes with highly conserved expression patterns ($p > 0.7$) across all three pairwise comparisons. The data underlying this figure can be found in https://zenodo.org/records/15016454.
(EPS)

**S6 Fig. Transcription factor fingerprints for central brain cell types.** A heatmap illustrating the percentage of cells expressing 104 transcription factor genes across 107 cell types, derived from the *D. melanogaster* dataset. Both genes and cell types are clustered through hierarchical clustering. Labels for glial cell types and Kenyon cell types are colored blue and red, respectively. The data underlying this figure can be found in https://zenodo.org/records/15016454.
(EPS)

**S7 Fig. Frequency comparisons across homologous cell types.** Bar plots showing comparisons across the drosophilid species of the frequencies of 107 central brain cell types. Each point corresponds to one of six replicate snRNA-seq datasets. The data underlying this figure can be found in https://zenodo.org/records/15016454.
(EPS)

**S8 Fig. *Tret1-1* expression levels in *Drosophila* species.** Dot plots summarizing the expression levels of *Tret1-1* across 107 cell types of central brains of *D. melanogaster*, *D. simulans*, and *D. sechellia*. The data underlying this figure can be found in https://zenodo.org/records/15016454.
(EPS)

**S9 Fig. Differences in glial cell types among three *Drosophila* species. (A)** Left, representative images of *Tret1-1* HCR FISH on *Dmel*CS, *Dsim*04, and *Dsec*07. Right, quantifications of *Tret1-1* signals for each species. Images and data were obtained from independent experiments from those shown in Fig 2D. **(B)** Left, representative images of Repo immunofluorescence on *Dmel*CS, *Dsim*04, and *Dsec*07. Right, quantifications of Repo+ cell counts for each species. Images and data were obtained from independent experiments from those shown in Fig 2F. The data underlying this figure can be found in https://zenodo.org/records/15016454.
(EPS)

**S10 Fig. Correlation between cell type frequencies and DEG counts.** Scatter plots illustrating percent of cell type frequencies (x-axis) and mean DEG counts from the three comparisons (*Dmel-Dsec*, *Dmel-Dsim,* and *Dsim-Dsec*) (y-axis). The data underlying this figure can be found in https://zenodo.org/records/15016454.
(EPS)

**S11 Fig. Predicted divergent gene expression in *D. sechellia*.** Dot plots illustrating expression levels and frequencies of 10 *D. sechellia*-specific DEGs in *D. melanogaster*, *D. simulans*, and *D. sechellia* across 107 cell types. The data underlying this figure can be found in https://zenodo.org/records/15016454.
(EPS)

**S12 Fig. In vivo validation of divergent gene expression in *D. sechellia*. (A)** Left, representative images of *CAH2* HCR FISH on *Dmel*CS, *Dsim*04, and *Dsec*07. Right, quantifications of *CAH2* signals for each species. Images and data were obtained from independent experiments distinct from those shown in Fig 5C. **(B)** Left, representative images of *ImpE1* HCR FISH on *Dmel*CS, *Dsim*04, and *Dsec*07. Right, quantifications of *ImpE1* signals for each species. Images and data were obtained from independent experiments distinct from those shown in Fig 5E. The data underlying this figure can be found in https://zenodo.org/records/15016454.
(EPS)

**S13 Fig. Impact of noni paste supplement on cell type frequencies in the *D. sechellia* central brain.** A dot plot showing comparisons of brain cell type frequencies between *D. sechellia* grown on standard medium (blue) and noni paste-supplemented medium (purple). Each point corresponds to one of six replicates. The error bars represent the standard error of the mean. A paired *t* test was conducted to assess statistical significance, with *P*-values adjusted using false discovery rate. No significant differences in cell type frequencies were observed between conditions. The data underlying this figure can be found in https://zenodo.org/records/15016454.
(EPS)

**S14 Fig. Expression changes of CG5151 in *D. sechellia* upon noni-paste supplement.** Dot plots summarizing the predicted expression changes of CG5151 across 107 cell types of brains of *D. sechellia* grown on standard food or with noni paste supplement. In the latter condition, CG5151 is upregulated in PRN and SUB, but this

increase is statistically significant only in PRN. The data underlying this figure can be found in https://zenodo.org/records/15016454.
(EPS)

**S15 Fig. In vivo validation of upregulation of CG5151 in *D. sechellia* upon noni-paste supplement** Left, representative images of CG5151 HCR FISH on *Dsec*07 grown under the indicated conditions. Right, quantifications of CG5151 signals. Images and data were obtained from independent experiments from those shown in Fig 6D. The data underlying this figure can be found in https://zenodo.org/records/15016454.
(EPS)

**S16 Fig. Changes in *D. sechellia* brain gene expression upon noni supplement.** Dot plots illustrating expression levels of four DEGs in the indicated cell types in response to noni treatment. The data underlying this figure can be found in https://zenodo.org/records/15016454.
(EPS)

## Code availability

All code for generating the figures and tables are available from Zenodo (https://zenodo.org/records/15016454).

## Acknowledgments

We thank the Flow Cytometry Facility (D. Labes) and Lausanne Genomic Technologies Facility (J. Marquis, C. Peter, R. Sermier, and K. Bojkowska) of the University of Lausanne for assistance with cell preparation and sequencing. We thank for B.-Y. Lee for bioinformatic support on single-cell analysis. We are grateful to members of the Benton laboratory for discussions and comments on the manuscript.

## Author contributions

**Conceptualization:** Daehan Lee, Richard Benton.

**Data curation:** Daehan Lee, Michael P. Shahandeh.

**Formal analysis:** Daehan Lee, Michael P. Shahandeh.

**Funding acquisition:** Daehan Lee, Richard Benton.

**Investigation:** Daehan Lee, Michael P. Shahandeh, Liliane Abuin.

**Methodology:** Daehan Lee, Michael P. Shahandeh.

**Project administration:** Daehan Lee, Richard Benton.

**Resources:** Daehan Lee, Richard Benton.

**Software:** Daehan Lee.

**Supervision:** Daehan Lee, Michael P. Shahandeh, Richard Benton.

**Validation:** Daehan Lee.

**Visualization:** Daehan Lee, Michael P. Shahandeh.

**Writing – original draft:** Daehan Lee, Richard Benton.

**Writing – review & editing:** Daehan Lee, Michael P. Shahandeh, Liliane Abuin, Richard Benton.

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
