## [Editor Report · Decision Letter 0]

9 Jan 2024

Dear Richard, 

Thank you for submitting your manuscript entitled "Comparative single-cell transcriptomic atlases reveal conserved and divergent features of drosophilid central brains" for consideration as a Research Article by PLOS Biology. Many thanks for your patience over the holiday period.

Your manuscript has now been evaluated by the PLOS Biology editorial staff, and I'm writing to let you know that we would like to send your submission out for external peer review. We have not been able to secure advice from an Academic Editor because the appropriate experts are currently all travelling or busy; in order not to delay your manuscript any further, I discussed your paper with the team, and we're happy to send it out for review.

Once your full submission is complete, your paper will undergo a series of checks in preparation for peer review. After your manuscript has passed the checks it will be sent out for review. To provide the metadata for your submission, please Login to Editorial Manager (https://www.editorialmanager.com/pbiology) within two working days, i.e. by Jan 11 2024 11:59PM.

Kind regards,

Roli

Roland Roberts, PhD

Senior Editor

PLOS Biology

rroberts@plos.org

---

## [Decision Letter · Decision Letter 1]

13 Mar 2024

Dear Richard,

Thank you for your patience while your manuscript "Comparative single-cell transcriptomic atlases reveal conserved and divergent features of drosophilid central brains" was peer-reviewed at PLOS Biology. It has now been evaluated by the PLOS Biology editors, an Academic Editor with relevant expertise, and by four independent reviewers. 

You'll see that reviewer #1 is very positive and only has a few textual requests, Reviewer #2 says that the work is of broad interest but questions the robustness of the algorithm used to analyse differences in cell composition, wonders if the differences in cell type frequencies could be validated with IHC or FISH, wants more clarity regarding cutoffs, and suggests that it might be better to map all genes to the same genome (Dmel). Reviewer #3 thinks this is a landmark study, but is concerned about the consequences of neutral processes (rather than adaptation) in D. seychellia, wants more support for the glial cell findings, and (like rev #2) questions the effects of mapping Dsec genes to the Dsim genome. Reviewer #4 (also like rev #2) wants you to validate the glial findings with IHC/FISH, and wants you to pay more attention to the unannotated clusters of genes.

I discussed these comments with the Academic Editor who said that, although they were usually reluctant to request additional experimental work, there was clear agreement among the reviewers - both in their actual reviews and in their cross-commenting - that the study would be substantially strengthened by the additional validation requested by reviewers #2 and #4.

In light of the reviews, which you will find at the end of this email, we would like to invite you to revise the work to thoroughly address the reviewers' reports.

Given the extent of revision needed, we cannot make a decision about publication until we have seen the revised manuscript and your response to the reviewers' comments. Your revised manuscript is likely to be sent for further evaluation by all or a subset of the reviewers.

**IMPORTANT - SUBMITTING YOUR REVISION**

*Re-submission Checklist*

*Published Peer Review*

*PLOS Data Policy*

*Blot and Gel Data Policy*

Sincerely,

Roli

Roland Roberts, PhD

Senior Editor

PLOS Biology

rroberts@plos.org

REVIEWERS' COMMENTS:

Reviewer #1:

This ms addresses the question of brain evolution with three species of fruit flies, the workhorse Dmel, Dsim and Dsech. These were carefully chosen to provide a good phylogenetic framework for interpretation, with the highlight on Dsech, a specialist to contrast with the two generalists. Dsim and Dsech are comparably evolutionarily diversed from Dmel in terms of time. Of course the work benefits immensely from the extremely well annotated Dmel genome.

The ms uses comparative snRNAseq as the one and only research tool, as a modern alternative to comparative neuroanatomy. 

Results are as predicted: more differences in Dsech relative to the other species. More surprising, perhaps is the finding of overall very high levels of similarity in sn gene expression across the species. Similarity is used to infer evolutionary conservation—see comment on this below.

A nice addition was to rear Dsech on standard medium and then on medium supplemented by its sole natural host; a few changes in gene expression were observed but mostly stasis. As this was the only manipulation performed in a study with a species (Dmel) that is manipulated more than it is described, this was noteworthy.

The methods are straightforward, using some of the best single cell RNAseq analytics approaches.

The paper is an information-dense descriptive analysis that should provide a foundation for follow up work on brain evolution. The authors are appropriately circumspect about causality vs. correlation. It would be great to see creative manipulations of gene expression in Dsim to see if a preference for Dsech could be induced.

Very scholarly treatment, engaging in some of the big discussion points in evolutionary neuroscience.

I have the following concerns—all relatively minor.

As with all studies of this young genre, there is an issue of how well can one classify cells based solely on expression patterns. As noted in an influential paper by David Schulz and colleagues, reliable classification occurs only with a combination of molecular and anatomical information, i.e., combining molecular cell type and anatomical cell type. This reviewer is not asking for a re-analysis, but it would be good to mention this concern as a caveat. 

Re "similarity" = conservation. Perhaps the standard lab environment is so limiting that species-specific differences are not induced. This certainly has been seen in considering the effects of genetic mutations on phenotype in mice, comparing standard impoverished lab environment vs an enriched environment. That is to say, the similarity = conservation conclusion is reasonable, but an alternative also exists that should be addressed.

The proportion of genes captured with this sn protocol appears lower than what one sees in other studies. If these are the most highly expressed or most stably expressed genes, they might be giving an overly conserved perspective.

In the Discussion, the opening implies that this study examined how brains evolve. I would argue that there is no "how" here, just snapshots of expression showing differences/similarities. That's different in how those came to be. That would require experimental evolution. That said, the present study certainly addresses this big and important question, providing key observations that could be the basis of experimental evolution studies.

Reviewer #2:

In this manuscript, Lee and Benton perform a comparative transcriptomic analysis of a trio of closely related drosophilids that have a very distinct phylogenetic and ecologic relationship: D. simulans (Dsim) and D. sechelia (Dsec), which are are sister species (diverged only ~0.2 million years ago) and D. melanogaster (Dmel), which diverged from the Dsim/Dsec clade ~4 million years ago. In terms of ecological niches, Dmel and Dsim are generalists, while Dsec is specialized on feeding on noni fruit. While the Benton lab has studied in the past the role of sensory neurons in this specialization, the authors here focus on the central nervous system and compare the neuronal composition of the central brain of these species.

To do this, they use single-nucleus mRNA sequencing to read the transcriptomes of ~45,000-50,000 cells from each drosophilid central brain coming from six independent biological replicates. After integrating these datasets, they found the overall composition of the central brains to be fairly conserved. They also identify a number of conserved gene expression profiles between cell types/clusters of the different drosophilids, which provide as a resource of cell type molecular fingerprints. They then address the more pertinent question, i.e. what are the differences in cell type composition in the drosophilid trio with a particular interest in how the noni specialist Dsec is different from the two generalists. In this, they are aided by the strategic selection of the species, which allows them to compare a duo of generalists (Dmel and Dsim) and a duo of a generalist and a specialist (Dmel and Dsec) that diverged at the same time (3-4 million years ago). Interestingly, most of the differences they find are in glial cells:

A) Dsec has proportionally more perineurial glial cells than Dmel and Dsim.

B) Four out of the five glial cell types show an accelerated rate of gene expression change in the noni specialist drosophilid (Dsec) than in the generalist (Dsim)

C) They find forty differentially expressed genes in the Dsec glial cell types, which is consistent with this increased divergence in gene expression

D) Finally, when they compare the transcriptome of Dsec central brain neurons in the presence or absence of noni juice paste in the food, they find four out of the six differentially expressed genes to be differentially expressed in glial cells. 

The above observations allow them to propose a model where the glial cells were the first to respond to the presence of noni fruit in their diet and they were subsequently evolutionarily reshaped to allow Dsec to adapt to this very specific niche.

Overall, it is a very interesting paper bringing forward a number of ideas that would not be able to be drawn from other experimental systems. It is of broad interest and it would be a very useful part of the literature. However, I have a number of comments that the authors could consider to make the conclusions of the paper more robust.

- I am still not very convinced by the use of single-cell sequencing data integration to identify differences in cell type composition. These algorithms are meant to integrate samples that are similar and may "force" different cell types to integrate. Did the authors try different algorithms or different parameters to integrate? How did they evaluate the quality of the integration? This information would be useful to be in the Methods. Moreover, I think that it would be useful for the authors to compare the three atlases without integrating. Do they find one-to-one correspondences between different dataset clusters based on marker gene expression?

- On a similar note, the difference in frequencies of homologous cell types should be better supported. First, there are different algorithms that try to evaluate such differences in a way that is tailored to single-cell sequencing data, such as miloR from the Marioni lab. Second, some of the differences the authors find could be a result of technical differences, e.g. in nuclear sorting. It would be worth if the authors could verify some of these differences in situ. For example, the difference in the number of perineurial glial cells should be fairly easy to capture by immunohistochemistry or FISH. 

Also, regarding the difference in frequencies, do the authors really believe that the number of clock neurons is affected when Dsec is grown with or without noni fruit?

- The authors use different cutoffs at different parts of the paper that are not justified. For example, in the section "Conserved gene expression patterns in drosophilid central brains" they select genes that are expressed in "at least 5% of Dmel cells". Then, they select genes that are expressed in over 30% of cells across species and focus on genes that are restricted to 1-9 clusters, etc. This is also observed in other parts of the paper. I am wondering how the authors came up with these cutoffs and whether the use of different cutoffs could have affected some of their conclusions.

- Finally, I am also concerned by the mapping of Dsec reads on the Dsim genome. This creates an imbalance between the two species (as Dsim is mapped in each own genome, while Dsec is mapped on a sister species genome). Since the entire study is based on the strategic selection of the drosophilid trio to dissect the differences in gene expression that are caused by adaptive versus neutral evolution, this imbalance may skew the results and the interpretation. Assuming that generating and annotating a reliable Dsec genome is challenging, wouldn't it be better that the authors do all their analyses with all the datasets mapped to the Dmel genome (i.e. Dmel-to-Dmel vs Dsim-to-Dmel vs Dsec-to-Dmel)?

Reviewer #3:

Lee & Benton use single-nuclei sequencing of the central brain of three drosophilids to present the first overview of brain evolution at the single cell level. The authors examine changes in cell type number and gene expression divergence between two generalist species and the feeding specialist D. sechellia. The authors find that cell types overall are highly conserved between these species, but that glial cell types on average exhibit greater between-species divergence than neuronal cell types. Divergence at the neuronal level is concentrated in two peptidergic cell types. Gene expression and cell type divergence occur faster on the branch leading to D. sechellia than to the generalist species D. simulans. This paper is a landmark in the field and is of broad interest. I have several suggestions to improve or strengthen the interpretations and analyses. 

Major suggestions:

A significant concern is the interpretation of the increased divergence in D. sechellia as adaptation to its niche. D. sechellia has a much smaller population size than D. simulans which is sufficient to explain the increased divergence without adaptation. The authors could specifically test for signatures of gene expression adaptation to support their adaptation conclusion. The authors could also note the alternative possibility of reduced purifying selection relative to mutation and drift which would increase divergence in small population sizes such as D. sechellia. Indeed, this could result in increased compensatory changes in gene expression, of which several were observed in D. sechellia in this study. 

One of the major conclusions of the paper is that glial gene expression diverges faster than that of neurons. This is very interesting and additional analyses would strengthen this conclusion. For example, is this analysis sensitive to the number of genes used (I.e. different than 50)? Furthermore, glial cells show larger cell type abundance than most neuronal clusters likely resulting in increased statistical power. It would be important to see whether the gene expression correlations are robust when cell types are subsampled to the same cell numbers. This finding would also be strengthened if the results extend to divergence in principal components and not just at the gene level. 

There should be some consideration of whether D. sechellia reads can be mapped to the genome of D. simulans without affecting the results. For example, more divergent genes may map less well to D. simulans giving the appearance of lower expression in D. sechellia. This may extend across multiple genes to give the impression of greater overall expression divergence in D. sechellia. 

Minor suggestions:

It seems notable that OPNs show the greatest gene expression divergence between D. melanogaster and the other two species. The authors may want to speculate on the significance of this given the importance of these neurons in the response to noni fruit and other odors. 

The authors state that 80 genes is too few for GO analysis. While there may be limited power with 80 genes, it does not preclude GO analysis. If no significant GO terms were identified with the 80 genes, this should be stated. 

"We focused on the 1,686 genes that are expressed in at least 5% of D. melanogaster central brain cells.": D. melanogaster Italic font.

"similar to its known role for enteroendocrine cell specification in the gut [49] Furthermore": add period after [49].

Reviewer #4:

This study by Lee and Benton generated snRNA-seq atlas data of the central brains from three closely related Drosophilids: Drosophila melanogaster, Drosophila simulans, and Drosophila sechellia. The data quality is good in terms of cell number, expressed genes, and replicates. In the cellular composition level, it was found that all major cell types are well conserved with a few outliers, such as perineurial glia, sNPF and Dh44 neurons. In the gene expression level, it was found that glial cell types showed the greatest divergence between species. 

Overall, this study provides a good resource for the fields of species evolution and comparative neural developmental biology, although there are several studies that have already profiled the central brain, whole brain, and the whole head from Drosophila melanogaster. My major two concerns include the lack of in vivo validation about their key findings on cellular differences and cell-types specific DEGs, as well as the lack of rigor on the clustering analysis. 

1. Validation. a) one of the major findings in this study is the identification three cell types in D. sechellia showing different ratios compared with other two species. Perineurial glial cells (PRNs) show a decrease and sNPF and Dh44 neurons show an increase. These ratio differences may come from sampling bias introduced by dissection especially with manual removal of optical lobes, by FACS, or by nucleus filtering steps during data analysis. Considering these factors, it is important to validate these findings with in vivo staining. b). when comparing DEGs among three species, it was found that four types of glial cells exhibited the highest number of DEGs, with unique gene expression changes in D. sechellia. These are interesting findings. Also, when comparing D. sechellia brains between flies raised in normal lab food and in lab food with noni, they found one gene CG5151 is induced by noni food. However, none of these findings are validated with any in vivo assays using RNA in situ, antibody staining, or genetic reporters, which are routine assays for the Benton lab. 

2. Clustering analysis. In Fig 1c, there is a clear D. melanogaster-specific cluster (red) in the center (green and blue tSNE plots showing a hole in the center), and there are also at least two blubs showing either more red cells (Dmel; top part) or more green/blue cells (Dsim/Dse; top and right). However, current analysis pipeline used by the authors could not detect them, which raise concerns for the overall clustering analysis. What are these clusters? What genes and pathways are enriched in these cells? Could these cells be identified as specific clusters with higher clustering resolution? I think these cells are assigned as unannotated (a very big group) so that the sub-type differences will not be revealed. Considering the fact that these unannotated cells take a large fraction (maybe >50% according to fig 1d), I suggest the authors need to pay more attention to them and do further analysis instead of ignoring them by assigning them as unannotated. 

Other comments: 

1. It is not justified or discussed why only female flies are used. 

2. When comparing DEGs across cell types, does the cell number affect DEG numbers? 

3. In the part of "Gene expression plasticity in the specialist brain", it is good to do and show the difference among the groups in Fig. 6a with statistical testing.

---

## [Decision Letter · Decision Letter 2]

4 Mar 2025

Dear Richard,

Thank you for your patience while we considered your revised manuscript "Comparative single-cell transcriptomic atlases reveal conserved and divergent features of drosophilid central brains" for publication as a Research Article at PLOS Biology. This revised version of your manuscript has been evaluated by the PLOS Biology editors, the Academic Editor and three of the original reviewers.

Based on the reviews, we are likely to accept this manuscript for publication, provided you satisfactorily address the following data and other policy-related requests:

IMPORTANT - please attend to the following:

a) Please could you make a slight change to the Title, to make it more appealing? We suggest: "Conserved and divergent features of drosophilid central brains suggest a role for glial gene expression changes in ecological adaptation" We lose the mention of scRNA-seq, which will nevertheless be apparent from the Abstract, but we introduce the "surprise" importance of glia (hopefully the verb "suggest" is soft enough?). Happy to discuss this further by email.

b) Please address my Data Policy requests below; specifically, we need you to supply the numerical values underlying Figs 1CDE, 2ABCDEF, 3ABCD, 4ABCD, 5ABCDEF, 6ABCDE, S1ABCD, S4, S5ABC, S6, S7, S8, S9AB, S10, S11, S12AB, S13, S14, S15, S16, either as a supplementary data file or as a permanent DOI’d deposition. I note that you already have an associated GitHub deposition (https://github.com/Evomics/FlyBrainEvo), which looks rather comprehensive. Please could you confirm whether this contains all the data and code needed to recreate the Figures? Also, because Github depositions can be readily changed or deleted, please make a permanent DOI’d copy (e.g. in Zenodo) and provide this URL (see below).

c) Please cite the location of the data clearly in all relevant main and supplementary Figure legends, e.g. “The data underlying this Figure can be found in https://zenodo.org/records/XXXXXXXX"

d) Please make any custom code available, either as a supplementary file or as part of your data deposition. I think that this is all in the Github deposition, but again, we need a Zenodo snapshot, and we need you to include the Zenodo URL in the data/code availability statement.

We expect to receive your revised manuscript within two weeks. 

*Published Peer Review History*

*Press*

Sincerely,

Roli

Roland Roberts, PhD

Senior Editor

rroberts@plos.org

PLOS Biology

DATA POLICY:

Regardless of the method selected, please ensure that you provide the individual numerical values that underlie the summary data displayed in the following figure panels as they are essential for readers to assess your analysis and to reproduce it: Figs 1CDE, 2ABCDEF, 3ABCD, 4ABCD, 5ABCDEF, 6ABCDE, S1ABCD, S4, S5ABC, S6, S7, S8, S9AB, S10, S11, S12AB, S13, S14, S15, S16. NOTE: the numerical data provided should include all replicates AND the way in which the plotted mean and errors were derived (it should not present only the mean/average values).

CODE POLICY

DATA NOT SHOWN?

REVIEWERS' COMMENTS:

Reviewer #2:

The authors have addressed my comments completely. Congratulations on a very interesting study.

Reviewer #3:

[identifies himself as Trevor Sorrells]

The authors have rigorously addressed all concerns raised. They should be commended on a valuable contribution to the field. 

Reviewer #4:

For the original manuscript, I had two major concerns about validation of key findings and the robustness of clustering analysis. In this revision, the authors validated the species-specific differences in cell types, and confirmed differential expression of glial genes. The authors clarified that the apparent species-specific clusters in the t-SNE plot were due to limitations of the visualization method. They introduced an improved analysis pipeline with iterative subclustering, which revealed 107 cell types.

These revisions enhance the robustness of the findings and improve the clarity of the clustering analysis. I have no more concerns for publishing this paper.

---

## [Editor Report · Decision Letter 3]

17 Mar 2025

Dear Richard,

Thank you for the submission of your revised Research Article "Comparative single-cell transcriptomic atlases of drosophilid brains suggest glial evolution during ecological adaptation" for publication in PLOS Biology. On behalf of my colleagues and the Academic Editor, Chris Jiggins, I'm pleased to say that we can in principle accept your manuscript for publication, provided you address any remaining formatting and reporting issues. These will be detailed in an email you should receive within 2-3 business days from our colleagues in the journal operations team; no action is required from you until then. Please note that we will not be able to formally accept your manuscript and schedule it for publication until you have completed any requested changes.

Sincerely, 

Roli

Senior Editor

PLOS Biology

rroberts@plos.org